# Machine learning-guided discovery of ionic polymer electrolytes for lithium metal batteries

Kai Li[1], Jifeng Wang[1], Yuanyuan Song[1] & Ying Wang [1] ✉

As essential components of ionic polymer electrolytes (IPEs), ionic liquids (ILs) with high ionic conductivity and wide electrochemical window are promising candidates to enable safe and high-energy-density lithium metal batteries (LMBs). Here, we describe a machine learning workflow embedded with quantum calculation and graph convolutional neural network to discover potential ILs for IPEs. By selecting subsets of the recommended ILs, combining with a rigid-rod polyelectrolyte and a lithium salt, we develop a series of thin (~50 μm) and robust (>200 MPa) IPE membranes. The Li|IPEs|Li cells exhibit ultrahigh critical-current-density (6 mA cm$^{-2}$) at 80 °C. The Li|IPEs|LiFePO$_4$ (10.3 mg cm$^{-2}$) cells deliver outstanding capacity retention in 350 cycles (>96% at 0.5C; >80% at 2C), fast charge/discharge capability (146 mAh g$^{-1}$ at 3C) and excellent efficiency (>99.92%). This performance is rarely reported by other single-layer polymer electrolytes without any flammable organics for LMBs.

Ionic polymer electrolytes (IPEs) containing the non-flammable ions embedded with the mechanically supporting polymers with pre-determined ionic pathway have received considerable attention toward reviving clean energy storage and conversion devices, such as batteries[1-6], fuel cells[7], supercapacitors[8], mechanical actuators[9] and reverse osmosis membranes[10]. As promising candidates for safe and environmentally friendly electrolyte materials, ionic liquids (ILs) are room temperature (RT) molten salts with low vapor pressure, high thermal stability, wide electrochemical window and high ionic conductivity[1,5,11]. In recent years, liquid crystalline polymers have shown the capability to effectively reduce interfacial resistance, meanwhile raising unique ion conduction mechanisms in lithium metal batteries (LMBs)[1,12]. Lithium (Li) metal anode coupled with high-energy-density cathodes, for example, Li-air and Li-sulfur batteries, usually require highly conductive, thermal-stable and electrochemical-stable electrolytes to suppress inhomogeneous Li dendrites, overcome the side reactions and break the tradeoffs between conductivity and modulus in the composite electrolytes[13,14]. To alleviate these issues synergistically, IPEs have shown the capability to block dendrites through the robust polymer matrix, meanwhile guaranteeing extreme safety by avoiding organic plasticizers in LMBs[13,15-17].

As critical components in IPEs, it is desirable to develop an approach to screen suitable ILs from a large population of IL candidates to develop successful IPEs for LMBs. Machine learning (ML) has been widely discussed to predict properties and learn the rules underlying datasets, thus efficiently simplifying the material-discovery process[18-23]. Here, we describe a ML workflow embedded with the quantum chemistry calculation and graph convolutional neural network (GCNN) to discover potential ILs with high ionic conductivity and sufficient electrochemical window. Driven by the structure-property relationships, previous researchers have developed diverse statistical methods and regression models to predict physical properties, for example, melting point[24], viscosity[25] and ionic conductivity[26] based on the structural descriptors of the ILs[27]. However, the reported high accuracy usually originates from the overfitting of the dataset[28]. Among the training datasets, the sample size of unique ILs is extremely limited[27,28]. The investigated datasets usually contain the datapoints of the same ILs at varying temperatures; these replicated datapoints will increase the appeal accuracy of the reported models artificially[25,26,29]. Thus, it is still challenging to predict the accurate properties of new ILs without enough labeled datapoints. To overcome the data scarcity issue, we comprehensively combine the object-oriented unsupervised

[1]Department of Macromolecular Science, State Key Laboratory of Molecular Engineering of Polymers, Fudan University, Shanghai 200438, China. ✉e-mail: wying@fudan.edu.cn

learning and supervised learning to emphasize the design of statistical regression and classification workflow instead of predicting the absolute physical properties of the IL pairs independently. In addition, this work also demonstrates the efficiency of using GCNN for the classification task based on the graph-to-property relationship of ILs.

Based on the screening results from ML, we experimentally investigate a series of IPEs based on the filtered ILs combined with a liquid crystalline polyelectrolyte Poly 2,2′-disulfonyl-4,4′-benzidine terephthalamide (PBDT) and a predetermined Li salt. The fabrication process includes two steps: First, the precursor composite membranes of PBDT with selected ILs were obtained by the solvent-evaporation method reported previously[30]. Then, the IPEs were finally achieved through the ion exchange step by immersing the composite membranes in highly concentrated ionic liquid electrolytes (ILEs). In terms of the polymer matrix, it has been demonstrated that PBDT can serve as the assembly template not only offering mechanical integrity and low interfacial resistance, but also endowing nanoscale structuring in the composite to ensure fast Li$^+$ transport[1,31,32]. In terms of the ion exchange medium, Li metal anode shows highly reversible cycling performance with a stable solid-electrolyte interphase (SEI) formed by electrochemical reduction of the bis(fluorosulfonyl)imide (FSI$^-$) anion[33,34]. In addition, compared to lithium tetrafluoroborate (LiBF$_4$) and lithium triflate (LiTfO), LiFSI shows excellent solubility in ILs with anion of FSI$^-$[35]. (The explanations about the solubility of different Li salts are summarized in Supplementary Note 1). In addition, N-propyl-N-methylpyrrolidinium bis(fluorosulfonyl)imide (C$_3$mpyrFSI) has a wide electrochemical window (5.4 V) and promising ionic conductivity (9.1 mS cm$^{-1}$)[35]. Thus, to ensure high Li$^+$ concentration in the IPEs after the ion exchange step, we employ LiFSI dissolved in C$_3$mpyrFSI with a concentration of 3.2 mol kg$^{-1}$ as the ion exchange medium to load Li$^+$ in the IPEs. By incorporating these IPEs into batteries with Li metal as an anode, we can experimentally confirm the properties of these developed IPEs, including ionic conductivity, Li$^+$ transference number, electrochemical window, and Li dendrite suppression.

## Results and discussion
### Machine learning-guided screening of ionic liquids
This ML workflow requires two main steps: (1) Unsupervised learning, followed by (2) Supervised learning to target promising ILs. As shown in Fig. 1, we obtain 74 cations and 30 anions from the web scrapping of the IoLiTec website. The permutation of cations and anions forms an IL candidate pool with 2220 unique ILs, but only less than 13% of the ILs show measured properties, for example, the melting point, viscosity, conductivity, and electrochemical window. Three open-source platforms, including RDKit, Psi4, and Pytorch Geometrics (PyG) are employed to generate the molecular descriptors of the raw dataset. RDKit is a powerful tool to calculate the molecular structure and three-dimensional (3D) descriptors of the molecules[36]. Psi4 is an open-source ab initio electronic structure program for high-throughput quantum chemistry[37]. We employ the self-consistence field (SCF) method with Hartee-Fork theory coupled with the basic set of 6–311 + G** to optimize the geometric structure and then calculate the energy, the highest occupied molecular orbital energy (E$_{HOMO}$), the lowest unoccupied molecular orbital energy (E$_{LUMO}$) and the molecular dipole moments for the cations and anions separately. In this work, we combine 60 molecular structure descriptors from RDKit for cations, anions, and cation-anion pairs and 14 calculated electronic structural variables from Psi4 for cations and anions in the final dataset. PyG is a library based on Pytorch to build and train GCNN for a wide range of prediction tasks. The unsupervised learning comprehensively utilizes boxplots, pair plots, and hierarchical clustering to summarize the fundamental rules underlying the dataset. In terms of supervised learning, we employ statistical regression and classification to promote screening efficiency. Among the 2220 ILs, we initially use the classification method to predict the phase (solid/liquid) of the cation-

anion pairs at RT. The predicted results are based on the ensemble learning of algorithms including support vector machine (SVM), random forest (RF), XGBoosting (XGB), and GCNN. Then we employ both statistical regression and classification to evaluate the conductivity of the ILs at 25 °C. These ILs are predicted to be liquid phase at RT in the previous classification step. To improve the screening efficiency, we classify the ILs into two categories, based on predetermined and tunable threshold of ionic conductivity value(σ). It is well known that the σ of solid-state electrolytes should be >1 mS cm$^{-1}$ to ensure the performance of electrolytes in real devices[38]. To guarantee the high conductivity of IPEs, the utilized ILs usually require a slightly higher σ > 5 mS cm$^{-1}$[1]. Therefore, 5 mS cm$^{-1}$ is the σ threshold used in this model. The last screening step is determined by the calculated electrochemical window value (ECW) based on the HOMO/LUMO theory according to Eqs. 1–3 through Psi4. The ECW of an IL is determined by both the cations and anions. The cathodic limit (V$_{CL}$) is the maximum value of the cathodic potential determined by the E$_{LUMO}$ for cations and anions. Similarly, the anodic limit (V$_{AL}$) is the minimum value of the anodic potential determined by the E$_{HOMO}$ for cations and anions[39]. For the ECW threshold, the LiFePO$_4$ cathode coupled with Li metal anode displays a charging platform at 3.5 V (vs V$_{Li+/Li}$). Based on the assumption of V$_{CL}$ ≥ V$_{Li+/Li}$, 3.5 V will be the minimum threshold for the ECW. If the ILs need to match other higher voltage cathodes, such as NMC811, LiCoO$_2$, and LiNiMn$_2$O$_4$, the threshold for the ECW can be adjusted appropriately. Here we set the threshold to 4 V in the workflow to ensure ubiquity. According to this ML workflow shown in Fig. 1, we finally obtain 49 ILs in the recommendation list. We will expand the discussion of unsupervised learning and supervised learning in the following sections.

$$V_{CL} = \max\left(-\frac{E_{LUMO}[+]}{e}, -\frac{E_{LUMO}[-]}{e}\right) \quad (1)$$

$$V_{AL} = \min\left(-\frac{E_{HOMO}[+]}{e}, -\frac{E_{HOMO}[-]}{e}\right) \quad (2)$$

$$ECW = V_{AL} - V_{CL} \quad (3)$$

### Unsupervised learning
To investigate the features and the underlying correlations in the dataset, we employ unsupervised learning based on boxplots, pair plots and hierarchical clustering. The boxplots shown in Supplementary Fig. 1a, b display the distribution of the σ of ILs according to the cation and anion types correspondingly. The ranking is based on the median value for each cation or anion type. Similarly, Supplementary Fig. 1c, d display the rankings for the measured ECW from IoLiTec of the ILs correspondingly. We observe that the ECW boxplots show large variations, but the ammonium-based cations and imide-based anions usually display promising ECWs. These boxplots are insightful for the initial selection of ILs in future investigations. In addition, pair plots are widely used to display the correlations between variables. Figure 2a shows the corelation between σ and viscosity of ILs, which can be elucidated by the Nernst-Einstein equation and Stokes-Einstein equation included in Supplementary Note 2[40]. However, as shown in Fig. 2b, there is no apparent correlation between σ and ECW. Thus, we conclude that σ and ECW of ILs are independent factors; meanwhile, σ and ECW are both essential properties for electrolytes as applied to LMBs. To validate the ECW calculated based on the HOMO/LUMO theory. We directly compare the calculated ECW with the results scrapped from IoLiTec. The bar plots shown in Fig. 2c, d display the average ECW and absolute difference between the results for different cation and anion types correspondingly. The mean absolute errors (MAE) between the calculated results and the experimental results are <1.1 V. As indicated

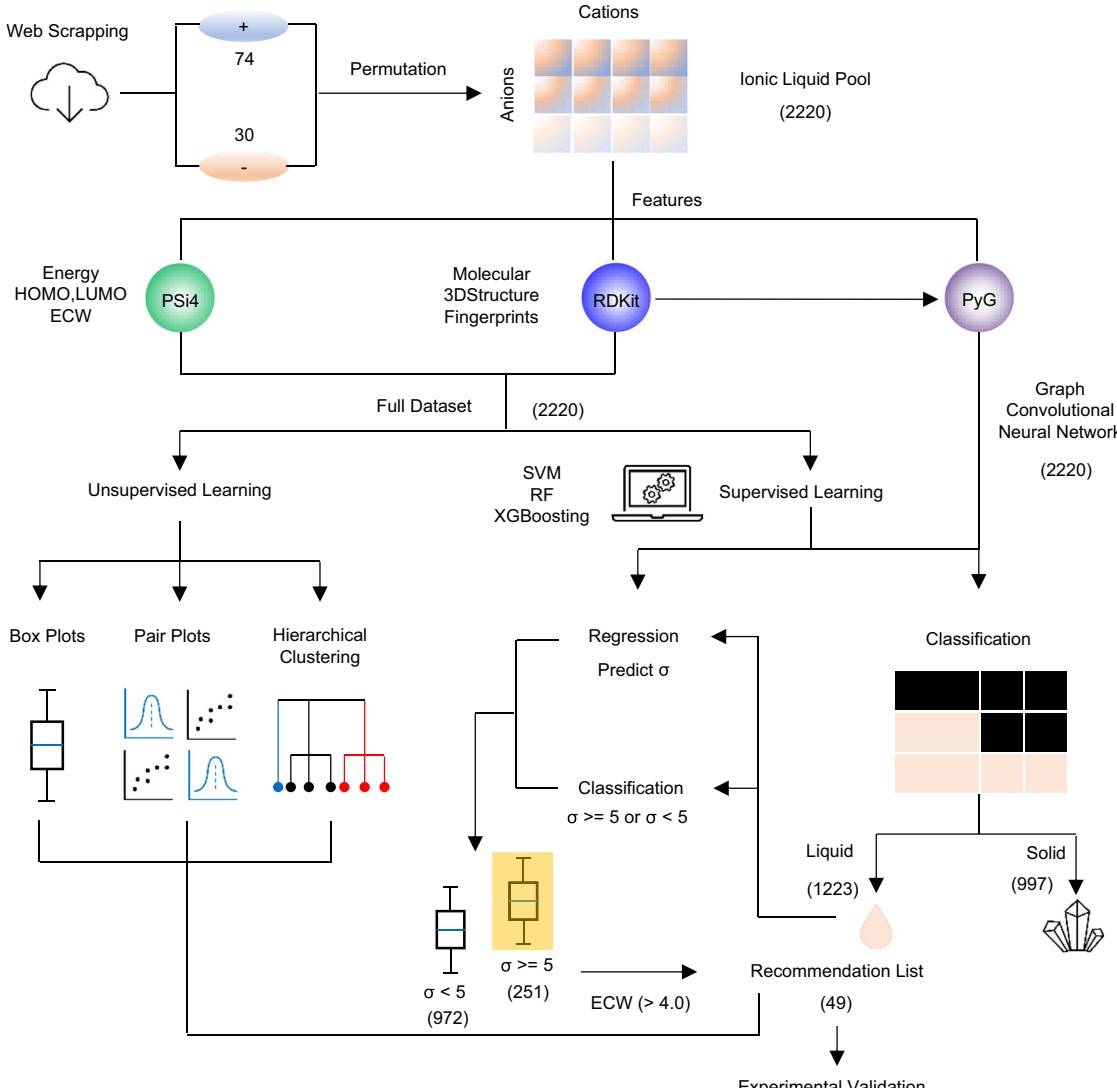

**Fig. 1 | Machine learning workflow for the discovery of ILs with high conductivity (σ) and wide electrochemical window (ECW).** The permutation of 74 cations and 30 anions forms an IL pool containing 2220 unique ILs. Three open-source platforms, including RDKit, Psi4, and PyG are employed to generate the molecular descriptors for the raw dataset. Unsupervised learning contains box-plots, pair plots, and hierarchical clustering, which are essential analytical methods for investigating the structure and correlations of variables in the dataset. Supervised learning leverages both regression and classification based on SVM, RF, XGBoosting and GCNN. The IL pool will initially be classified as a solid or liquid group. Then the ILs with liquid phase at RT will be further classified based on the σ ≥ 5 mS cm⁻¹ or not. Meanwhile, we also employ regression to predict the absolute σ values of the ILs for reference. Finally, ECW > 4 V is the final screening criterion for the final recommendation list of potential ILs.

in the literature, it is still challenging to estimate ECW for ILs accurately[39,41]. We observe that the derivations for some cation and anion types are higher. The explanation for the uncertainty in groups like imidazolium and BF$_4$ is included in Supplementary Note 3. In addition, the measured ECW values are highly dependent on experimental conditions, thus we believe that the MAE achieved by the calculated ECW is overall satisfactory and of significant reference to the field. Last, hierarchical clustering has recently been employed to predict fast inorganic Li ion conductors[42]. Here, we also utilize this algorithm to detect the clustering of ILs with high conductivity along with wide ECW. The key features used for the hierarchical clustering include the computed ECW and the top 15 features of the XGB model (when classifying the conductivity type, the specific features are included in Supplementary Note 4). The final hierarchical clustering dendrogram is displayed in Fig. 2e. Additionally, in order to validate the effectiveness of the clustering, we also plot the finally screened ILs (49) based on the supervised learning in Fig. 2f. We can clearly observe that the

results based on the two learning protocols are highly overlapped, which indicates the high efficiency of the unsupervised learning as compared to the supervised learning introduced in the following section for material screening and discovery.

## Supervised learning

Building on unsupervised learning, we also propose a multistep supervised learning to filter ILs with desired σ and ECW at RT. First, based on the ensemble learning of SVM, RF, XGB, and GCNN, we predict the phase (liquid or solid) of the ILs in the pool. Supplementary Fig. 2. displays the heatmap for the phase prediction results in the permutation table between cations and anions. In Fig. 3a, we divide the predicted results into four categories, including liquid and solid-x/3, where x (x = 1, 2, 3) is the number of ML models (SVM, RF, XGB) with prediction results being in solid phase for the ILs. The larger the number of x, the higher possibility for the IL being in a solid phase. To further validate the predicted results, we employ quantum chemistry

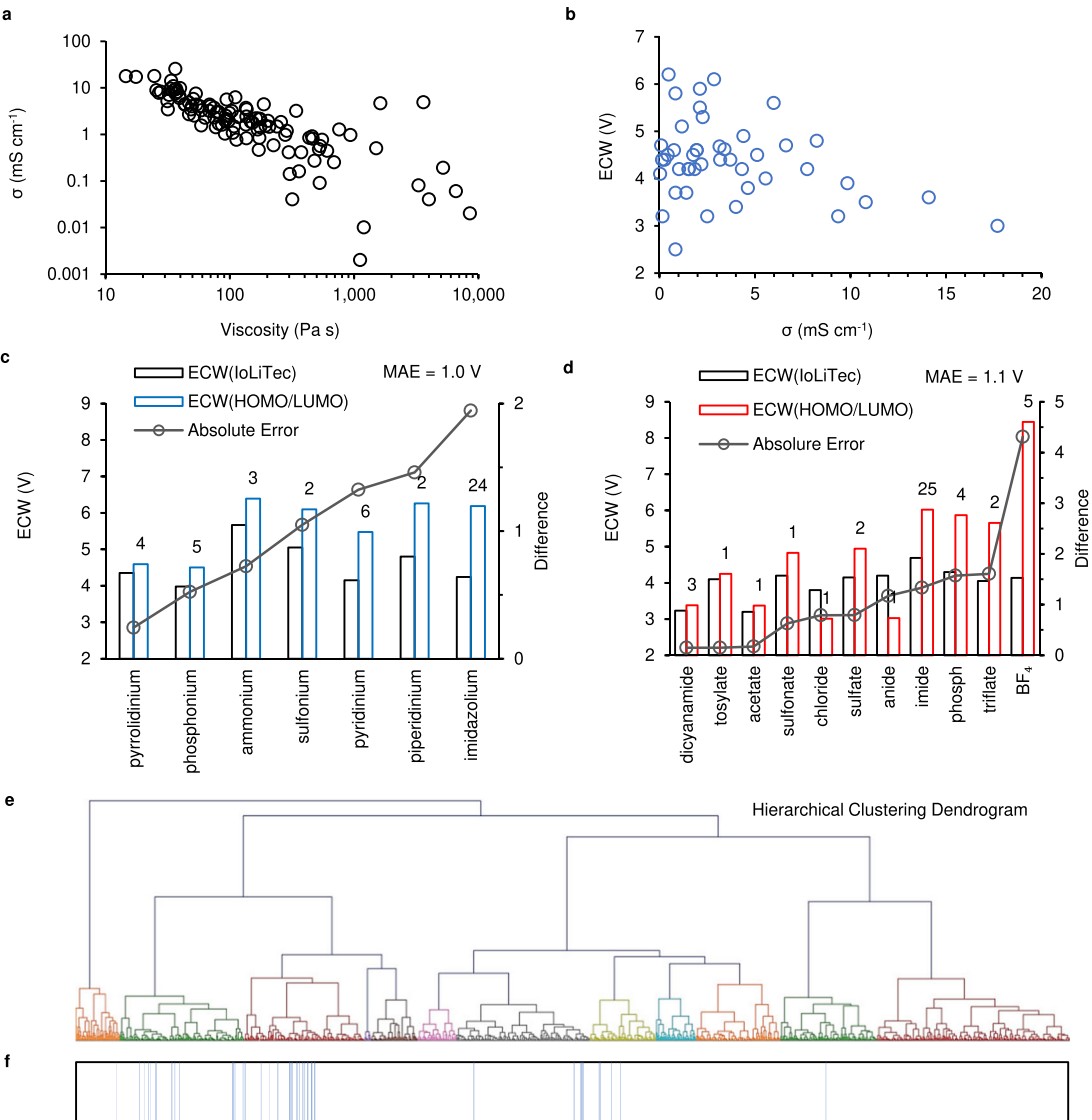

**Fig. 2 | Unsupervised learning of the dataset. a** The relationship between σ and viscosity of the ILs with known properties in the dataset. **b** The relationship between the σ and the ECW of the ILs with known properties in the dataset. **c**, **d** Comparison of ECW based on IoLiTec to ECW based on HOMO/LUMO theory for the cation (**c**) and anion (**d**) types, correspondingly. The mean absolute error

(MAE) is also indicated. **e** Hierarchical clustering dendrogram for the ILs based on the top 15 features (when classifying the conductivity type of the ILs in the supervised learning) and the calculated ECWs. **f** The blue lines show the screened ILs based on the supervised, which is highly consistent with the clustering results achieved by the unsupervised learning shown in (**e**).

calculation to calculate the binding energy ($E_{binding}$) between cations and anions. The $E_{binding}$ is calculated through Eq. 4, where $E_{opt}$ refers to the energy of the optimized geometry for specific cations ($E_{opt[+]}$), anions ($E_{opt[-]}$) or dimers ($E_{opt[+][-]}$). The lower of the $E_{binding}$ means easier for the cations and anions to stay in tightly associated pairs and pack into solid crystals[43]. We select 91 ion pairs, including representative 7 cations and 13 anions from the main cation and anion types in the dataset. Among the 91 ion pairs, there are 19 ILs with known phases from IoLiTec. The corresponding average $E_{binding}$ for the labeled solid and liquid groups are shown in Fig. 3a. The $E_{binding}$ ($\sim -400\,kJ\,mol^{-1}$) of the solid group is indeed lower than the liquid group. In terms of the predicted results for the remaining 72 ILs, the liquid group shows the highest $E_{binding}$; meanwhile, we really observe that the $E_{binding}$ decreases as x increases, not only confirms our demonstration that the solid cation-anion pairs usually show lower $E_{binding}$, but also validates the efficiency of this ML classification to separate liquid/solid candidates at RT. Detailed calculation results are included in Supplementary Tables 1 and 2. To validate statistically, we

perform both one-way ANOVA and T-test to validate the difference between the liquid group as compared to the other three solid groups. Both of the hypothesis testing results indicate significant differences as shown in Supplementary Tables 3 and 4. The T-test results show more details and indicate significant differences between the liquid and Solid-2/3, Solid-3/3 except for Solid-1/3. Thus, we can conclude that there is a significant difference between the liquid group and groups with more than 2 models showing solid prediction results. To discover the key features, we start with the features indicated by the feature importance score of the model one by one, finally, we observe that the sphericity index of cation and anions are correlated with the phase of ILs. As shown in Fig. 3b, the phase of ILs is highly dependent on the geometric structures of the cations and anions. Thus, we also employ the geometric GCNN model to investigate this classification. The details about the GCNN model used in this work are included in Supplementary Note 5. Additionally, the $E_{LUMO}$ of the anion seems like another important feature to determine the phase of the ILs, which deserves further investigation.

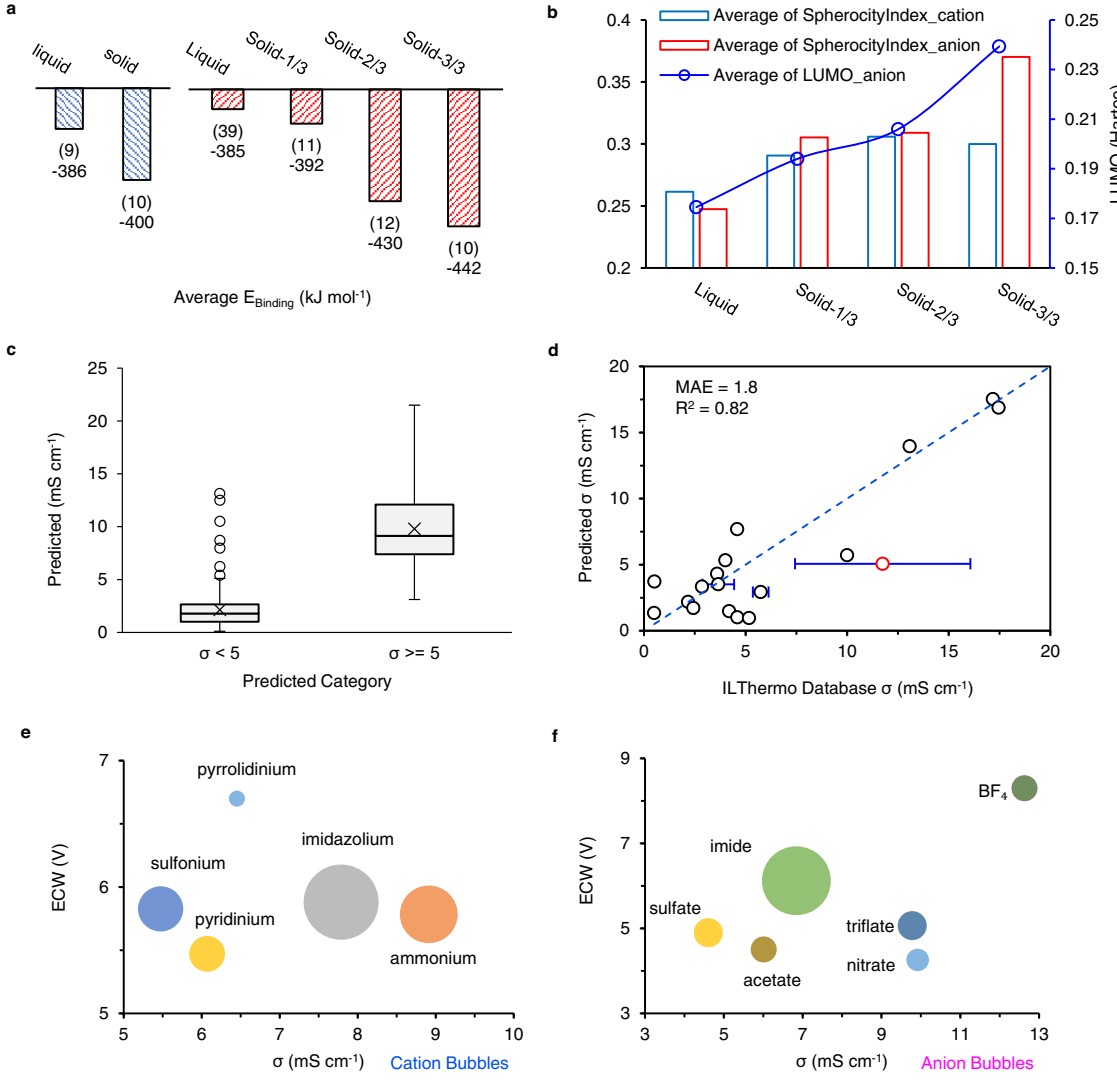

**Fig. 3 | Supervised learning of the dataset. a** The column plots for the average binding energy ($E_{binding}$) between cations and anions in specific groups. Blue columns show the average $E_{binding}$ for the ILs with known phases. Red columns show the average $E_{binding}$ for the ILs with predicted phases, including liquid and solid-x/3, where x (x = 1, 2, 3) is the number of ML models (SVM, RF, XGB) with prediction results being in solid phase for the ILs. The column labels refer to the number of ILs and the average $E_{binding}$ in each group. **b** The key features when classifying solid/liquid phases of the candidates and their average values for each group. **c** The boxplots of predicted σ for the two categories with σ ≥ 5 mS cm$^{-1}$ or not. **d** Comparison of the predicted σ in the test dataset to literature reported σ in the ILThermo Database at 25 °C. The horizontal error bar shows the standard deviation of the experimental values stored in ILThermo database. The red circle is excluded in validation for its substantial uncertainty. **e, f** The bubble plots show the relationship between ECW and σ of the final recommended ILs grouped by cation and anion types, separately. The bubble size refers to the count of ILs in each category.

After we obtain liquid ILs candidates based on the previous solid/liquid classification, we further classify the conductivity type (σ ≥ 5 or σ < 5) of the ILs based on the known σ in the dataset. The performance of the employed models is included in Table 1. To eliminate the overfitting issue, the reported accuracy for SVM, RF, and XGB algorithms are the average of the 5-fold cross-validation accuracy results. We observe that the GCNN model offers similar performance only based on the geometric descriptors of cations and anions without any additional calculation results, which shows high promise for this algorithm for future materials genome projects. The GCNN is not applied for the conductivity classification and prediction tasks to simplify the complexity of the ML workflow.

Next, we apply the other three models to predict the absolute σ of the liquid ILs at RT. The boxplots shown in Fig. 3c further indicate the consistency between the regression and the classification results. The median value of the predicted conductivity for the group with σ < 5 is 1.8 mS cm$^{-1}$. Meanwhile, the median value for the group with σ ≥ 5 is

9.1 mS cm$^{-1}$. Finally, we propose 49 ILs with σ ≥ 5 mS cm$^{-1}$ along with a calculated ECW > 4 V in the recommendation list for developing IPEs in LMBs. The full list is included in Supplementary Table 5. To validate our prediction results, we compare the predicted σ of the liquid ILs in this work to the experimental results stored in the NIST ILthermo Database[44]. The ILthermo Database contains 523 unique ILs with measured σ values at varying temperatures. As shown in Fig. 3d, though only 18 ILs are overlapped between our predicted σ and ILThermo database σ at 25 °C (298.15 K), we observe high consistency between the predicted results and the ILthermo results, especially for those with high σ. The $R^2$ factor is 0.82, and the mean absolute error (MAE) is 1.8 mS cm$^{-1}$. More discussions about the $R^2$ based on this distinctive validation are included in Supplementary Note 6 and Supplementary Table 6. It is also difficult for us to find similar $R^2$ validation results in the literature, which compares the unique ILs with an external database at a single temperature as we reported here. Overall, we conclude that this model can perform well in predicting the σ of ILs

**Table 1 | The performance of the classification tasks based on different algorithms**

| Algorithm | Accuracy* (Solid or Liquid) | Accuracy* ($\sigma \geq 5$ or $\sigma < 5$) |
|---|---|---|
| Support Vector Machine (SVM) | 0.81 | 0.85 |
| Random Forest (RF) | 0.83 | 0.80 |
| XGBoosting (XGB) | 0.85 | 0.82 |
| Graph Convolutional Neural Network (GCNN) | 0.83 | NA |

*The accuracy score for SVM, RF, and XGB algorithms indicate the average of the 5-fold cross-validation accuracy results.

with high $\sigma$. The bubble plots shown in Fig. 3e, f indicate the distribution of the filtered ILs based on the cation and anion types, correspondingly. In terms of the cations, we observe that the ammonium and imidazolium cations display the highest average $\sigma$ that agrees with the box plots shown in the unsupervised learning section. Regarding the anions, $BF_4$, triflate, and imide show promising ECW and $\sigma$. Based on these preliminary rules, we select 5 ILs from the bubbles and conduct the experimental validation in the following section.

$$E_{binding} = E_{opt[+][-]} - E_{opt[+]} - E_{opt[-]} \qquad (4)$$

### Electrochemical performance of IPEs based on the filtered IL

To verify the efficiency of this screening process, we develop a series of IPEs based on 5 ILs on the final recommendation list and validate the cycling behavior and electrochemical performance of the developed IPEs coupled with the Li metal anodes. The selected ILs include 1-ethyl-3-methylimidazolium triflate ($C_2$mimTFO), 1-ethyl-3-methylimidazolium tetrafluoroborate ($C_2$mimBF$_4$), 1-ethyl-3-methylimidazolium ethyl sulfate ($C_2$mimES), diethylmethylammonium triflate (DemaTFO) and diethylmethylsulfonium bis(trifluoromethuldulfonyl)imide (DemsTFSI). All these ILs show both high $\sigma$ and wide ECW at RT. The fabricated IPEs are transparent and mechanically strong with 5/10 wt% of PBDT. We employ the stress-strain tests for the IPEs with 10% PBDT and $C_2$mimTfO. As shown in Supplementary Fig. 3a, the yield strength of the membrane is 6.21 MPa. The Young's modulus based on the slope of the linear portion on the curve is ~300 MPa, which is 3000 times higher than that of PEO-based electrolyte (0.1 MPa)[45,46]. In Supplementary Fig. 3b, the dynamic mechanical analysis results show that this membrane maintains a high modulus >200 MPa from −50 to 300 °C, which ensures the safety and thermal stability of this material as applied to real devices. After ion exchange with the ionic liquid electrolyte (ILE) with LiFSI dissolved in $C_3$mpyrFSI, we obtain a large area and flexible IPE membranes. We also employ NMR and DSC to confirm the extremely low amount of $H_2O$ in the IPEs, which guarantees the excellent performance of IPEs in LMBs (Supplementary Fig. 4). In Fig. 4a, we perform cyclic voltammetry (CV) on Li||IPEs|SUS to evaluate the Li plating (negative scan) and stripping (positive scan) behavior of the selected 5 IPEs. We observe that upon scanning in the negative direction, only IPE with DemaTFO displays no Li deposition in the cathodic deposition, all other 4 IPEs display excellent cathodic stability along with high current density that offers promise for these IPEs as conductive electrolytes in LMBs. The ion exchange protocol fails to incorporate Li ions in DemaTFO-based IPEs, which indicates that the ion association behavior and varying binding energies are essential factors in determining the ion transport in the IPEs. In the following discussion, we will only focus on the other 4 IPEs. Overall, these 4 IPEs show promising ECW approaching 6 V. For the positive direction, as shown in the enlarged curves in the insets, we observe that $C_2$mimTFO and DemsTFSI-based IPEs show the highest anodic stability as compared to $C_2$mimBF$_4$ and $C_2$mimES. This ensures the

excellent full cell performance of IPEs based on $C_2$mimTFO and DemsTFSI as indicated in the following section. As reported previously, the calculated anodic limits for $BF_4^-$ as shown in Fig. 2d are usually overestimated when using the vacuum calculation model compared to the experimental results[39]. The reasons for the inconsistency originate from many factors, for example, the assumption of the calculation model and the complicated ion association in the real system. For comparison, we include the CV results for the neat ILs in Supplementary Fig. 5. Surprisingly, the cathodic stability for the neat ILs is much lower compared to the IPEs. We propose that the high $Li^+$ concentration, the decomposition of $FSI^-$ and the liquid crystalline PBDT in the IPEs can comprehensively improve the cathodic stability of the composite electrolytes, which is analogy to the "water in salt" electrolytes proposed by Suo et al.[47]. In addition to the SEI formed by the decomposition of $FSI^-$, the solvation and coordination numbers of the $Li^+$ will change dramatically with the adjustment of the relative cation and anion concentrations. We have demonstrated that the rigid-rod polyelectrolyte PBDT backbone will selectively absorb cations and anions during the ion exchange process, which will promote the Li transference number ($t_{Li}^+$) of the system. Here, we estimate the $t_{Li}^+$ based on the steady-state current of the Li symmetric cell assembled with IPEs in Fig. 4b. Supplementary Note 7 includes the Bruce-Vincent analysis and shows the corresponding impedance spectra of the cells before and after the polarization[48]. As shown in Supplementary Table 7, the determined $t_{Li}^+$ in these IPEs (0.4–0.5) are much higher compared to the pure ILE with LiFSI|$C_3$mpyrFSI ($t_{Li}^+$ = 0.18)[49]. Among these 4 IPEs, $C_2$mimTFO and DemsTFSI-based IPEs show higher $t_{Li}^+$ = 0.5 compared to $C_2$mimBF$_4$ (0.4) and Dema Ethyl Sulfate (0.4) for the varying interaction between PBDT chains and the anions in ILs. We are conducting in-depth quantum chemistry calculations to measure the interaction energies between anions and PBDT polymer chains, thus offering a clearer idea about the determining factors in the ion exchange process. The thickness of the electrolytes (~50 μm) is determined by SEM as shown in Fig. 4c. Figure 4d shows ionic conductivities of the IPEs as a function of temperature. The exceedingly high $\sigma$ at RT (~1 mS cm$^{-1}$) originates from the fibrillar and nanocrystalline conducting phase formed in the composite structure of IL and liquid-crystalline polymers reported previously[1,6]. Meanwhile, all samples show stable Li stripping and plating at varying current grades in the Li||Li cell cycling process. As shown in Fig. 4e, $C_2$mimTFO, $C_2$mim ES and DemsTFSI display stable cycling at high current density ($J$) at 6 mA cm$^{-2}$ at 80 °C, which is promising, since most organic cells reported cannot sustain any stable performance at this high temperature without any safety concerns.

In this section, we mainly extend the investigation of IPEs based on $C_2$mimTFO at varying current densities and cell configurations. The IPEs based on DemsTFSI show promising performance as well, the detailed results for DemsTFSI are included in Supplementary Fig. 6. In Fig. 5a, we initially test the symmetric cell performance at varying $J$ at RT. The critical $J$ is 2.0 mA cm$^{-2}$ at RT. As shown in Fig. 5b, the cells can maintain at least 800 h at 1 mA cm$^{-2}$ at RT without a short circuit, which is rarely seen by other single-layer polymer electrolytes without any supporting separators or organic plastiziers[50]. In addition, we prepare the Li|IPEs|Cu cell to investigate the plating and striping of Li at the Cu anode based on these IPEs. Figure 5c shows the voltage-cycle profile for the cells at RT. The secondary axis shows the corresponding coulombic efficiency (CE) values, the average value of which is >98%. The reported average CE for Li||Cu cells using regular organic electrolytes is ~90%, which indicates the highly reversible Li deposition on the Cu surface of these IPEs[51–53]. The SEM image of the deposited Li on the current collector is included in Supplementary Fig. 7. We observe no Li dendrites formed using these IPEs with Li metal anodes, which is believed to originate from the high modulus of the IPEs.

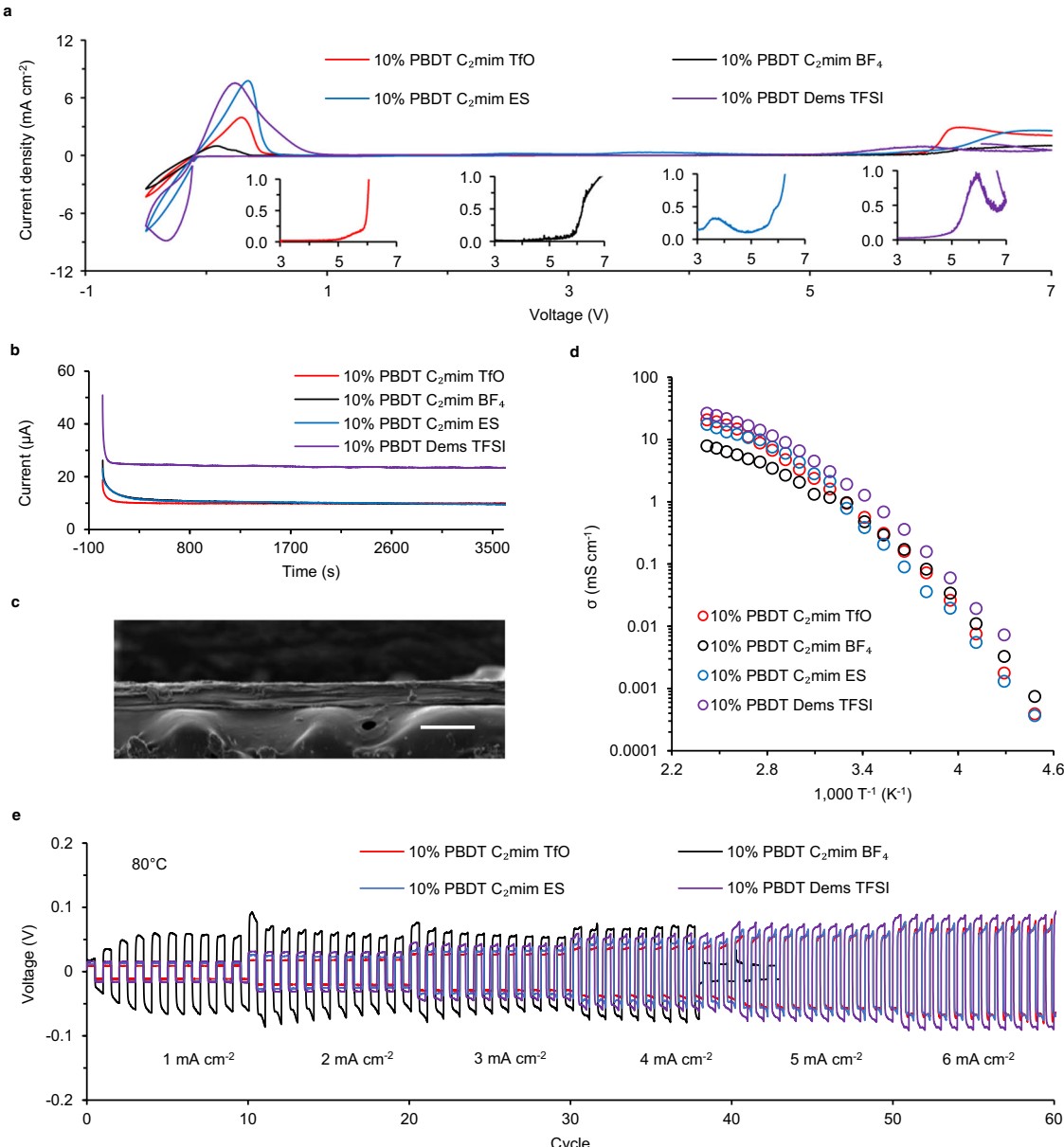

**Fig. 4 | Electrochemical window, Li⁺ transference number, ionic conductivity(σ) and Li symmetric cell cycling performance in IPEs. a** Cyclic voltammetry curves at a sweep rate of 10 mV s⁻¹ in IPEs developed with C₂mim TFO, C₂mim BF₄, C₂mim ES and Dems TFSI. Inset plots show the enlarged view of the cycles in high voltage range. **b** Steady-state current in symmetric Li||Li cell under 10 mV polarization.

**c** The thickness of the IPEs (~50 μm) is characterized by SEM. The scale bar is 100 μm. **d** Arrhenius plots for σ of the 4 IPEs. **e** Cell voltage versus time for a symmetric Li||Li cell assembled with the 4 IPEs at current densities (*J*) from 1 to 6 mA·cm⁻² with changes in *J* every 10 cycles at 80 °C (each cycle lasts 1 h).

For practical application, we also report the full cell performance of C₂mimTfO-based IPEs using the LiFePO₄ cathode with high loading (10.3 mg cm⁻²). Figure 6a shows the long-term cycling of the full cell at RT at 0.5 C rate, the cell shows 96% capacity retention at 350 cycle. The voltage capacity files for the selected main cycles are shown in Fig. 6b. The major population of the CE is between 100.1%–100.2%, which is slightly higher than the theoretical maximum value (100%) for the thermal fluctuation at the ambient temperature. Thus indicates the high reversibility of the Li metal cells based on this IPEs at RT. Figure 6c shows the long-term cycling of the full cell at 50 °C at 2 C rate, the cell shows not only a high average CE (>99.9%), but also 80% capacity retention at the 350th cycle, which is promising to satisfy the fast charge/discharge requirements for widely used portable devices. The voltage capacity files for the selected main cycles are shown in Fig. 6d. At last, we also investigate the fast charge/discharge capability and

thermal stability of these IPEs at 80 °C, which is a dangerous temperature for regular organic cells. Figure 6e shows the cycling performance of the IPE at an increasing C rate from 0.5 C (0.53 mA/cm²) to 5 C (5.3 mA cm⁻²) at 80 °C. Here, we observe that the cell maintains high capacity (~120 mAh g⁻¹) without short circuit at super high *J* up to 5 C (8.3 mA cm⁻²), which show promise for these IPEs as next-generation solid-state electrolytes for fast charge/discharge LMBs at mid-high temperatures. The corresponding voltage capacity curves for the selected cycles are shown in Fig. 6f. We further conclude a comparison to recent literature as shown in Table 2[12,54–59]. Overall, the IPEs reported in this work outperform from a comprehensive perspective, including the current density, the cell cycling life, and especially the high cathode loading required for practical applications.

In summary, we have described a ML-guided screening protocol to filter promising ILs with high ionic conductivity and wide

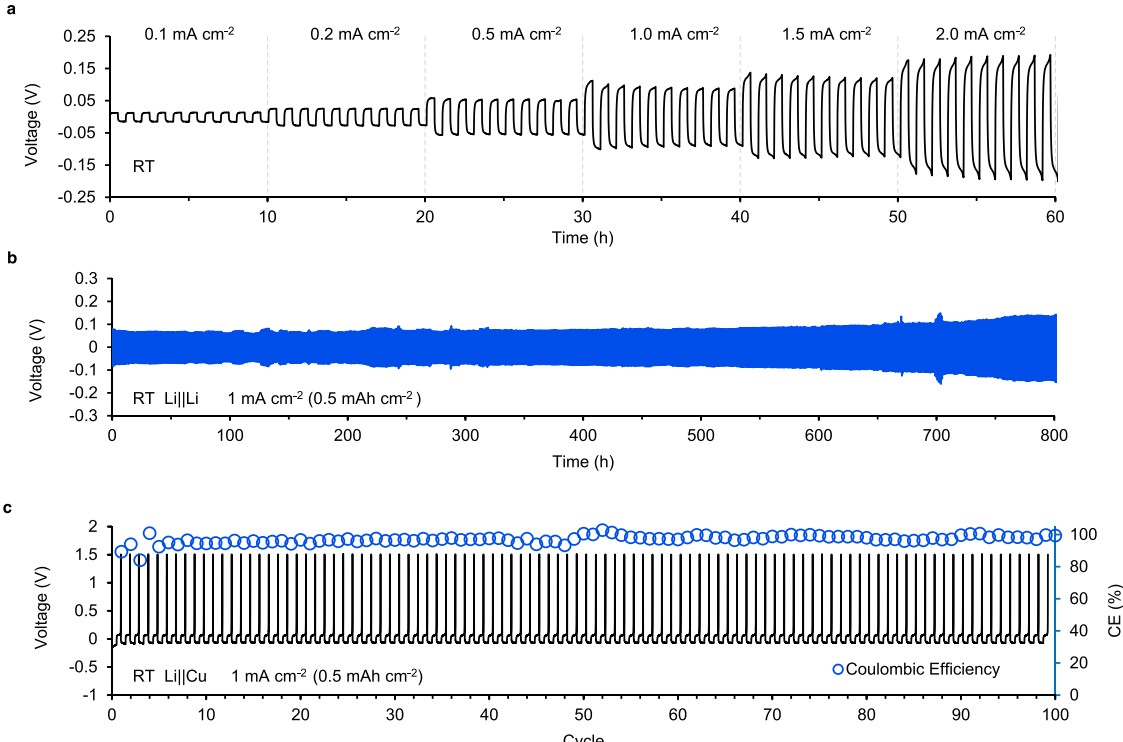

**Fig. 5 | Critical current density in Li‖Li and coulombic efficiency in Li‖Cu cells.** **a** Voltage-time profile for Li|IPEs|Li cell at room temperature (RT) with increasing *J* from 0.1 to 2.0 mA cm$^{-2}$. **b** Long-term voltage-time profile for Li|IPEs|Li cell cycled with 1 mA cm$^{-2}$ (0.5 mAh cm$^{-2}$) at RT. **c** Voltage-cycle profile for Li|IPEs|Cu cell cycled with 1 mA cm$^{-2}$ (0.5 mAh cm$^{-2}$) at RT. The secondary axis shows the corresponding CE values with an average value >98%.

electrochemical window for the preparation of IPEs in LMBs. In terms of the ML model, through the unique object-oriented unsupervised learning and multistep supervised learning. This comprehensive approach is essential to improve the efficiency to target promising ILs for practical applications. Compared to previous literature, instead of focusing on individual properties, for example, melting point, viscosity, and ionic conductivity, we first combine the factor of ionic conductivity with the electrochemical window as the guidelines for the selection of battery electrolytes. This novel conceptual design is also insightful and can be easily applied to related research areas. In addition, though there are plenty of published works based on ML and ionic liquids, it is still difficult to predict the ionic conductivity for ILs accurately because of the data scarcity issues. This work focuses on unique and commercially available cations and anions from IoLiTec company instead of the widely used and scattered NIST ILThermo database. This helps the research work better align with the commercially available products, we believe this is also significant for practical research and new materials design in the future. In terms of the electrolyte material development and performance evaluation, the promising experimental results reported in this work represent the performance of the state-of-the-art polymer electrolytes for Li metal batteries. We further confirm the rigid-rod liquid crystalline polyelectrolyte PBDT as an essential polymer matrix to develop a series of solid-state polymer electrolytes with extremely high CE and excellent fast charge and discharge performance at high temperatures. PBDT rods can serve as the assembly template not only offering mechanical integrity but also endowing nanoscale structuring in the composite, ensuring the fast Li$^+$ transportation. Overall, this platform shows immense potential to serve as an efficient method to quickly focus on the essential ILs for specific applications. More importantly, this work provides novel insights into strategies to overcome data scarcity issues and realize the efficient utilization of ML in material design and optimization. Through investigation of the golden rules, we could fabricate IPEs with tunable variations in mechanical, structural, and transport properties for a large array of applications in versatile functional devices, including batteries, fuel cells, supercapacitors, mechanical actuators and so on.

## Methods
### Machine Learning
Python programming language was used to conduct this machine-learning workflow. For the unsupervised learning, Dendrogram function from the SciPy package was used to perform the agglomerative hierarchical clustering. For the supervised learning, three main algorithms, including SVM, RF and XGB from the Scikit-learn package were used for the regression and classification tasks. The raw input data were web scrapped from the IoLiTec company website. The 74 input features for the cations, anions, and the permutations were calculated from RDKit (60) and Psi4 (14). Default features based on rdkit.Chem.Descriptors (10) module and rdkit.Chem.Descriptors3D (10) module of the cation, anion, and cation-anion pair were obtained by RDKit. These two modules are representative and contain detailed molecular and geometric properties of the molecules. The cation, anion, and cation-anion pair were modeled with the same set of descriptors based on RDkit. The remaining 14 features based on Psi4 will be introduced in the quantum chemistry calculation as below. GridSearch function was employed to obtain the optimized parameters for the three models. A 5-fold cross-validation was used to analyze the performance of the models during the training step for SVM, RF, and XGB. The NIST ILThermo Database was used to validate the predicted conductivity. Graph convolutional neural network was employed to conduct the classification in the supervised learning according to the RDkit and PyTorch Geometric (PyG) library. The machine learning workflow mainly consists of 10 sections, the code was packaged in a Class object named "ILP". All the code and the description for the codes are available on the GitHub website with links included in the Data Availability section.

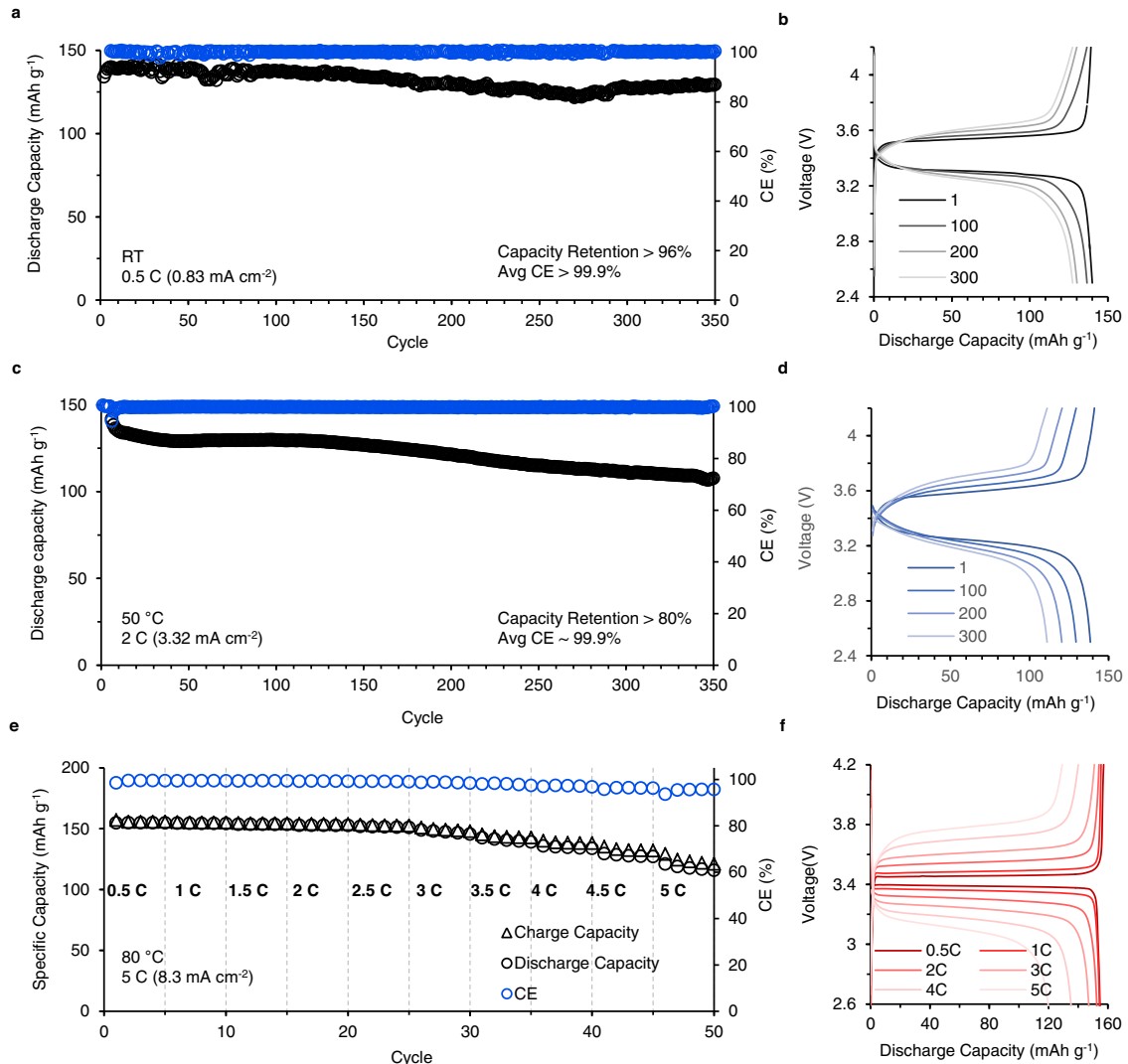

**Fig. 6 | Long-term cycling performance of Li|IPEs-C₂mimTfO|LiFePO₄ cell at varying temperatures and C rates. a** Cycling performance of Li|IPEs|LiFePO₄ cell at 0.5 C (0.83 mA cm⁻²) at RT. The black circles show the specific discharge capacity as a function of the increasing cycle number. The blue circles display the coulombic efficiency (CE) for each cycle correspondingly. **b** The voltage-capacity profiles for the main cycles in (**a**). **c** Cycling performance of Li|IPEs|LiFePO₄ at 2 C (3.32 mA cm⁻²) at 50 °C. **d** The voltage-capacity profiles for the main cycles in (**c**). **e** The Li|IPEs|LiFePO₄ cell cycling profile at elevated rates from 0.5 C to 5 C (8.3 mA cm⁻²) at 80 °C confirms the thermal stability when using this electrolyte at high temperatures. **f** The voltage-capacity curves for the main cycles are shown in (**e**).

**Table 2 | Comparison of the LMB cells based on IPEs in this work to recent literature**

| Materials | Temperature (°C) | Cathode Loading (mg cm⁻²) | Current density (mA cm⁻²) | Cycle number | References |
|---|---|---|---|---|---|
| LiFePO₄- FMC-ASPE-Li | 70 | 1–2 | 0.07–0.14 | 400 | (2022) Nat. Commun[54]. |
| LiFePO₄- FEC-SPE -Li | 22 | 12 | 0.36 | 60 | (2022) Nat. Nano[55]. |
| LiFePO₄- PI/PEO -Li | 40 | 6.9 | 0.08 | 60 | (2019) Nat. Nano[56]. |
| LiFePO₄- Li-Cu-CNF -Li | RT | 4.5–7.5 | 0.15 | 200 | (2021) Nature[12] |
| LiFePO₄- PEO/LSTZ -Li | 45 | 3.5 | 0.15 | 350 | (2019) PNAS[57] |
| LiFePO₄- CPE-05MC -Li | 55 | 5 | 0.5 | 100 | (2021) JACS[58] |
| LiFePO₄- HSE-EMIM-PP13 -Li | RT | 6.5 | 0.18 | 50 | (2022) Nat. Nano[59]. |
| LiFePO₄- IPEs -Li | RT | 10.3 | 0.83 | 350 | This work |
| LiFePO₄- IPEs -Li | 50 | 10.3 | 3.32 | 350 | This work |

## Quantum chemistry calculation

The quantum chemistry calculation was performed using Psi4 based on the self-consistence field (Hartee-Fork) method with basis set of 6−311+g** for the calculation of the energy, the HOMO, the LUMO, the total dipole moment, and the individual dipole moments along the three main directions for the cations and anions in the IL permutation table. The Psikit module, which is an ensemble of RDkit and Psi4, was used to connect and perform the calculation. In terms of the calculation of binding energies between the cations and anions, the commercially available Gaussian 16 software was employed to optimize the geometrical structure of the cations and anions separately, then the cation-anion pairs were further optimized based on the initially optimized cations and anions. The geometry optimization and frequency check were completed with B3LYP/6−311 g**. The energy calculations were based on M062X/6−311 + G(2d, p). The dispersion correction of DFT-D3 was applied for both optimization and single-point calculation. The convergence criteria were set by the Gaussian software with default values.

## Materials

PBDT was synthesized by interfacial condensation polymerization of 2,2′-Benzidinedisulfonic acid (BDSA) and Terephthaloyl chloride (TPC). BDSA (with 30 wt% water, Alfa Aesar) was purified through recrystallization before use. TPC (99%, Sigma Aldrich) was purified by vacuum sublimation prior to use. Polyethylene glycol with a molecular weight of $200 \, g \, mol^{-1}$ (PEG 200, General-Reagent), solvent chloroform (99.5%, General-Reagent), and catalyst sodium carbonate (>99.95%, Adamas-beta) were used without further purification. $C_2mimTfO$, $C_2mimBF_4$, $C_2mimES$, and DemaTFSI were purchased from IoLiTec (>99%) and used without further purification. $C_3mpyrFSI$ (purity > 99.9%) and LiFSI (purity > 99.5%) were purchased from Solvionic and used without further purification. Dimethylformamide (DMF) was sourced from Adamas. Lithium metal (Li-metal) foil with a thickness of 180 μm was sourced from Hongwei Lithium Co. Ltd. (purity > 99.9%). All ILs were further dried in a vacuum before moving to the glove box. The $LiFePO_4$ cathode with an area loading of $10.3 \, mg \, cm^{-2}$ and Cu foil were sourced from Canrd New Energy Technology Co., Ltd.

## Preparation of the ionic polymer electrolytes (IPEs)

0.45 g ILs and 0.05 g PBDT were dissolved in 5 g distilled $H_2O$ separately. Note, the only exception is that the solvent for DemaTFSI is DMF. The two solutions were mixed while heated up to 50 °C. Then the mixture solutions were poured into an 80 mm × 60 mm PTFE mode. The water/DMF solvent was slowly evaporated at RT and followed by vacuum drying at 80 °C for at least 24 h before transferring to the glove box. The membrane was then cut into round disks with a diameter of 9 mm and then immersed in the ionic liquid electrolytes (ILEs) for >24 h at RT. The ILEs were prepared by adding the prescribed amounts of LiFSI to $C_3mpyrFSI$ IL at RT in an Ar-filled glove box (<0.01 ppm $O_2$ and <0.01 ppm $H_2O$). The ILE was LiFSI dissolved in $C_3mpyrFSI$ with a concentration of $3.2 \, mol \, kg^{-1}$.

## Symmetric and full cells

The Li-metal symmetric coin cells were prepared with CR2032 cases with two 5 mm diameter Li-metal electrodes and 9 mm diameter IPEs (~50 μm) between them in an Ar-filled glove box. The Li|IPEs|Li symmetric coin cells were used for polarization, impedance spectroscopy, and cycling measurements. The Li|IPEs|LiFePO4 and Li|IPEs|Cu coin cells were also prepared for investigation of the electrochemical cycling performance, correspondingly. The thickness/diameter dimension of the working electrodes of $LiFePO_4$ is 80 μm/4 mm and Cu is 10 μm/4 mm. A NeWare Technology system was used for battery testing.

## Cyclic voltammetry (CV)

A stainless steel working electrode and a Li-metal foil counter electrode were employed for CV. The thickness of the Li metal counter electrode used in the CV experiment is 180 μm and the diameter is 5 mm. The CV measurements were performed against $Li|Li^+$ redox potential. All scans were performed at RT with $10 \, mV \, s^{-1}$ scan rate using a Biologic VMP 3e controlled by EC-Lab (ver. 10.40) software.

## Ionic conductivity

The ionic conductivity was measured via dielectric response over a 100 mHz−1 MHz frequency range for the assemble coin cells with stainless steel as the electrodes. A temperature scan range of −50 °C to 150 °C was selected, and the temperature was controlled by an oven. The ionic conductivities of the IPEs were obtained by fitting the electrochemical impedance spectra to an equivalent circuit model using EC-Lab (ver. 13.40) software®.

## Li transference number ($t_{Li}^+$)

The transference number was determined by direct current (DC) polarization. An AC impedance test was first performed over a 1 Hz to 1 MHz range to obtain the interfacial resistance before and after the polarization. The Li symmetric cells were polarized at ambient temperature with a constant potential of 10 mV for 1 h to obtain a stable current.

## Morphological and mechanical characterizations

Scanning electron microscopy was performed using a Zeiss Gemini SEM500 FESEM equipped with a Leica EM VCT500 microscope. Dynamic mechanical analysis was performed with TA Q800 using the stress-strain test of IPEs at a stress rate of $2 \, N \, min^{-1}$ under nitrogen atmosphere at RT. The slope of the stress−strain curves at <0.5% strain yields the Young's modulus. The tension mode was used to determine the storage modulus of IPEs at 1 Hz frequency. The test sample was first cooled to −50 °C and then heated to 300 °C at a heating rate of $2 °C \, min^{-1}$.

# Data availability

All data generated and analyzed in this study are included in this published article and its supplementary information files and are also available at https://github.com/wangyingxie/ILP.

# Code availability

The codes for the machine learning workflow, Psi4 calculation and GCNN predictions are also available online at https://github.com/wangyingxie/ILP with https://doi.org/10.5281/zenodo.7932384[60].

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

## Acknowledgements

This work was supported primarily by the China National Natural Science Fund for Excellent Yong Scientists Fund Program and Shanghai Pujiang Talents with award number 21PJD003.

## Author contributions

Y.W. conceived ideas, designed ML steps and composed and edited article drafts. K.L. executed and analyzed all the main electrochemical experiments and edited article drafts. J.W. helped with the quantum chemistry calculation, ML code organization, and edited article drafts. Y.S. helped with the execution of the electrochemical window experiments and edited article drafts.

## Competing interests

The authors declare no competing interests.
