## [Peer Review File · Nature Communications]

REVIEWER COMMENTS

Reviewer #1 (Remarks to the Author):

This manuscript offers a novel and combinatorial approach using data science and machine learning for prediction and classification of ionic liquids with optimised properties, which are then used as property enhancers in solid state electrolytes for lithium-metal batteries. Although the concept behind this work is promising and widely applicable, I personally believe that this research needs further work before reaching the publication stage, even if this is a Communication and not a full Research Article. If this Communication is to be published, I would recommend that, at a minimum, the comments below be addressed.

- What are the noteworthy results?

The strong advantage of this work is the combination of computational and experimental studies. It is very common in the Materials Chemistry field for the researchers to focus either on strictly theoretical or experimental studies, and a work showcasing the complementarity of the two is an excellent idea. The development of ionic polymer electrolytes incorporating ionic liquids as enhancers of their mechanical or physicochemical processes is currently a hotspot of scientific interests, with a plethora of papers being published every day. However, the authors of this work have managed to create an algorithm which can predict and classify the ionic liquids with the desired set of properties (something which is directly expandable to other research fields), but also actually create a set of fuel cells with very promising behaviour.

- Will the work be of significance to the field and related fields? How does it compare to the established literature? If the work is not original, please provide relevant references.

While the idea behind the simultaneous classification/prediction machine learning algorithm is brilliant, I am not convinced that this work will be of great significance for the field. The main reason for this is the absence of adequate validation of their predictions, which makes then their classification ambiguous. The authors are using a machine learning algorithm to predict the properties of interest in their system (here conductivity and electrochemical window), which however is at no point compared to literature data for the same compounds. I understand that since the studied dataset comes as a result of ion combinations, some of those ionic liquids are not commercially available, or maybe they have never been reported at all. But a comparison on a selected subset of ionic liquids is crucial.

Regarding the experimental data, from a pool of 40 recommended ionic liquids the authors select 4 to use in their fuel cells. However, I believe that the selection of ionic liquids is not in agreement with the scope of the work. All 4 selected ionic liquids are commercially available and I believe that they were chosen so that the authors avoid synthesising their own compounds. The whole point of

this work is to discover an optimised selection of ions for this application, but then the authors decide to work with 1-ethyl-3-methylimidazolium triflate and tetrafluoroborate, which are 'old' ionic liquids and they have been used since the 1990's for electrochemical applications. I believe that as a proof of concept they should have tried at least one 'unconventional' combination of ions from their list. The results they obtain on their fuel cells are indeed very promising, but for an expert on the ionic liquid electrochemistry field it is not surprising or unexpected.

- Does the work support the conclusions and claims, or is additional evidence needed?

The experimental investigation is thorough and their results well-explained. The prediction and classification algorithm part though needs further work and validation in order to prove that the algorithm works properly. As I mentioned in the previous section, there is no cross-reference for the prediction accuracy of the calculated properties. This should definitely be amended before publishing this article.

- Are there any flaws in the data analysis, interpretation and conclusions? - Do these prohibit publication or require revision?

During the 'supervised learning' section the authors introduce an additional filter to validate their data, by calculating the interaction energy between the anions and the cations. Figure 3c and Supplementary Table 1 show the results of these Energy calculations. To my opinion these calculations are problematic and they need revisiting. According to their formula, the energy values should be negative because the energy of the ion pair ($E[+][-]$) should be more negative than the individual energies of the ions. However, all the energies that are shown in Supplementary Table 1 are positive. The values of $E[+][-]$, $E+$ and $E-$ should also be shown in the ESI, for comparison reasons. Moreover, from my experience, the scale of those energies is unexpected, as I would expect the interaction energies in the range of minus a few hundreds kJ/mol. Depending on the geometry optimisation process the interaction energy can differ, so the graph shown in Figure 3c is expected to have an error bar of ± 50 kJ/mol, which makes it impossible to extract certain results from it, as the two box plots mostly overlap. Before the article is considered for publication, the authors need to revisit their calculations, explain properly their calculation methodology and answer questions such as what convergence criteria they use, specify grid accuracy, did they check for absence of imaginary frequencies etc?

Finally, something that is not so much of a flaw in the data analysis but more a part that needs further clarification is the part about the ion exchange with the ionic liquid electrolyte (rows 232-235). It is not clear to me why the incorporation of Li to the IPE is done with LiFSI and not with Li salt with the corresponding anion as the IPE (LiOTf and LiBF₄ accordingly) and also why the ion exchange happens in [C3mpyr][FSI]? I assume that the authors have tried different solvents and have a reason

for performing the ion exchange in another IL and not in a conventional volatile molecular solvent, but this should be clearly explained in the text.

- Is the methodology sound? Does the work meet the expected standards in your field?

Generally, the methodology of the work follows a reasonable pipeline, although some gaps exist (they are extensively discussed in the previous sections). These gaps need revising before considering this work for publication.

I am not convinced where the novelty and the importance of this work lies. There are many published articles on both ML algorithms for property prediction and conductive ionogels. I think that the authors need to add a paragraph (probably in the conclusions) highlighting the novelty of their work, what are the advantages of their algorithm compared to others and why their methodology/IPEs is superior to others reported in the literature.

- Is there enough detail provided in the methods for the work to be reproduced?

Yes, apart from the quantum chemistry calculations discussed above, the rest of the methodology is adequately explained and can be reproduced.

Regarding the ML algorithm, and while not being an expert on the field, I believe that the algorithm is sound and reasonable. I am not convinced that one could reproduce the algorithm just by reading the article's methodology section, since there are several gaps on the actual coding, but since the code is available on GitHub one could use the pre-made code.

- General Comments

1. There are several grammatical/syntax errors throughout the text, language needs to be corrected.
2. The company selling the ionic liquids is IoLiTec (in the text it is written as IoLiTech).
3. The authors refer to their setup as full cell in several points in the text. I believe it is supposed to be fuel cell.
4. The authors need to amend their abbreviations for their ionic liquids. For example 1-ethyl-3-methylimidazolium triflate, here it is abbreviated as C2mimTFO, the established abbreviation for that is either [Emim][OTf] or [C2C1im][OTf]. The authors need to correct their ILs abbreviations according to the literature.
5. Figure 4. The embedded photos are too small for anyone to see them properly.

6. Row 52. 'Large population of ion pairs' is wrong. It should be a large population of IL candidates
7. Row 52. Here they use the abbreviation ML but they define it later in text.
8. Row 181. Geometric not gyometric.
9. Row 232. If the authors claim that their films are mechanically strong, they need to perform a tensile stress analysis.
10. Row 350. The Machine Learning methodology needs to be more detailed in order to be more reproducible.
11. Row 360. The quantum chemistry calculations methodology needs more details to be reproducible.
12. Rows 365-373. The purity/water content of the purchased ionic liquids needs to be stated. Is Canrd in row 373 correct?
13. Rows 389. The thickness/diameters of the electrodes need to be stated.
14. Row 534. The github link is not working.

Reviewer #2 (Remarks to the Author):

In this manuscript, machine learning approaches are applied to discover ionic liquids with high ionic conductivity and electrochemical window. Some of the top candidates are blended with a polymer and a Li salt. Experiments are conducted to assess the performance of the ionic polymer electrolyte. Given the tremendous interest in Li-ion batteries and favorable attributes of ionic liquids as electrolytes, the topic of the manuscript has broad appeal. Developing electrolytes based on ionic liquids requires careful tuning of properties such as ionic conductivity, electrochemical window, viscosity etc., which are probed in this manuscript. Furthermore, machine learning approaches are becoming mainstream in the field of ionic liquids, so the manuscript holds special interest.

Although the topic of the manuscript is of great importance, there are several aspects of the manuscripts which deserve attention and substantial revisions.

- Very few details on the development of machine learning models are provided. I also find the development of the three machine models rather problematic as the data set is not split into a training and test data set making it difficult to assess the predictive capability of the model. This is

especially concerning as it has been pointed out in the manuscript that the models tend to overfit due to over-representation of one or more cation/anion types.

- RDKit generates a large number of molecular descriptors for a given ionic liquid. Please include a description of how the total number of features was reduced to 60. Also specify if the cation and anion in an ionic liquid pair were modeled with different set of descriptors.

- Provide a reasoning for the basis set selection. Please include the level of theory for electronic structure calculations,

- Why was a particular threshold of ionic conductivity and ECW selected?

- It is confusing that output properties from models such as ionic conductivity and electrochemical windows are listed as features.

- It is not clear that spherical anions will yield liquids/solid from Figure 3a. In fact, there does not seem to be a correlation between the sphericity of the anions and phase.

- How were top 10 important features determined?

- The results on 20 ILs is not enough to establish the correlation. For 1000 ILs, it is not difficult to conduct quantum calculations. I would recommend at least 20% of the ILs on which these calculations are carried out ensuring that cation types and anions are well represented.

- Why is it necessary to choose hydrophilic ionic liquids for Li-ion batteries? In fact, this property will actually be detrimental to the performance of batteries if water is absorbed.

- Provide the rationale for using PBDT as a liquid crystalline polyelectrolyte.

- The Github link is not provided.

- Provide a caption for Supplementary Table 2. Include units for ECW and ionic conductivity. Specify the temperature at which ionic conductivity is predicted. For many of the ionic liquids, measured ionic conductivities are available from the NIST ILThermo Database. A comparison must be made between predictions and the measurements.

Response to Reviewer #1's Comments

Statement: This manuscript offers a novel and combinatorial approach using data science and machine learning for prediction and classification of ionic liquids with optimised properties, which are then used as property enhancers in solid state electrolytes for lithium-metal batteries. Although the concept behind this work is promising and widely applicable, I personally believe that this research needs further work before reaching the publication stage, even if this is a Communication and not a full Research Article. If this Communication is to be published, I would recommend that, at a minimum, the comments below be addressed.

Response: We thank the reviewer for recognizing the potential importance of this work. The comments proposed by the reviewer are significant and insightful. Thus, we hope that the substantial revision to this manuscript based on the comments will be satisfactory.

Comment #1: What are the noteworthy results? The strong advantage of this work is the combination of computational and experimental studies. It is very common in the Materials Chemistry field for the researchers to focus either on strictly theoretical or experimental studies, and a work showcasing the complementarity of the two is an excellent idea. The development of ionic polymer electrolytes incorporating ionic liquids as enhancers of their mechanical or physicochemical processes is currently a hotspot of scientific interests, with a plethora of papers being published every day. However, the authors of this work have managed to create an algorithm which can predict and classify the ionic liquids with the desired set of properties (something which is directly expandable to other research fields), but also actually create a set of fuel cells with very promising behaviour.

Response: Thanks again to the reviewer for recognizing the uniqueness of our work. In terms of the noteworthy results of this work, the developed machine learning protocol not only relieves the issue of data scarcity, but also confirms the importance of machine learning in materials design and optimization. In addition, this work also emphasizes the development of materials from experimental perspective; meanwhile, proposing great potential of the developed IPEs in functional devices. For example, the assembled LMBs using IPEs coupled with commercial LiFePO_4 cathode (with high loading 10.3mg cm^{-2}) and bare Li metal anode deliver outstanding capacity retention for > 350 cycles ($> 96\%$ with 0.5 C at RT; $> 80\%$ with 2 C at $50\text{ }^\circ\text{C}$), fast charge/discharge

capability (146 mAh g⁻¹ with 3 C at 80 ° C) and ultrahigh coulombic efficiency (> 99.92%). This performance is rarely reported by any single-layer polymer electrolytes without any organic plasticizers/oligomers for LMBs. (Please refer to Comment #2 for more details about the performance compared to the state-of-the-art literature).

More importantly, this work provides an open-source machine learning model that can be simply reproduced and modified to screen ILs with characteristic parameters for related research areas. The implemented open-source tools, for example RDKit, Psi4, Pytorch and Pytorch Geometrics are all freely available. Through the high-throughput calculation, we can build a collection of databases that serves as an indispensable component in applying ML to materials science.

Comment #2: Will the work be of significance to the field and related fields? How does it compare to the established literature? If the work is not original, please provide relevant references.

Response: We appreciate the reviewer for raising these concerns. We will illustrate the significance and compare this work to the literatures from two perspectives, including the machine learning model and the experimental performance, correspondingly.

In terms of the machine learning model, this work is original and complementary. First, as mentioned by the reviewer, there is plenty of published work based on ML and ionic liquids. However, it is still difficult to predict the ionic conductivity for ILs accurately because of the data scarcity issues. Most of the published work is based on the ILthermo Database, which contains properties for diverse IL based on previous literatures.^{1,2} However, the dataset usually shows highly scattered datapoints and contains duplicate datapoints for individual IL. Especially when the model is mainly based on the geometric structure of the cations and anions, datapoints for the same ILs at varying temperatures will artificially increase the training efficiency of the model and cause the issue of overfitting.^{1,2} **In this work, we are only focusing on unique and commercially available cations and anions from IoLiTec company instead of widely used and scattered NIST ILThermo database.** This helps the research work better aligning with the commercially available products, we believe this is also significant for practical research and new materials design in the future. **The second point we would like to emphasize is the unique design of this screening steps based on a combination of unsupervised learning and multistep supervised learning.** This is essential to improve the efficiency to target promising ILs for practical applications. Compared to previous literatures, instead of focusing on individual

properties and predicting the absolute physical properties of the IL pairs purely, for example, melting point³, viscosity⁴ and ionic conductivity², **we first combine the factor of ionic conductivity with the electrochemical window, which are critical properties for battery electrolytes.** This novel conceptual design is also insightful and can be easily applied for related research areas.

In terms of the material development and performance evaluation, the promising experimental results reported in this work represent the performance of the state-of-the-art polymer electrolytes for Li metal batteries. We conclude a comparison to recent literatures [1-7] as shown in **Table R1. Overall, the IPEs reported in this work outperform from a comprehensive perspective, including the current density, the cell cycling life and especially the high cathode loading required for practical applications.**

In summary, this work is original in terms of the machine learning model, the quantum chemistry computational approach, the design of the materials and the excellent performance achieved in LMBs.

Modifications to the manuscript and supplementary information:

- (1) We added a paragraph (*Page 25, Line 409 – 431*) in the end of the manuscript to emphasize the novelty and importance of this work from the two perspectives.
- (2) We inserted Table R1 in the main manuscript as Table 2.

Table R1. Comparison of the LMB cells in this work to recent literatures.

Materials	Testing temp (°C)	Cathode Loading (mg cm ⁻²)	Current density (mA cm ⁻²)	Cycle number	References
LiFePO4- FMC-ASPE-Li-Li	70	1-2	0.07-0.14	400	1 (2022) Nature communications ⁵
LiFePO4- FEC-SPE - Li	22	12	0.36	60	2 (2022) Nature Nanotechnology ⁶
LiFePO4- PI/PEO -Li	40	6.9	0.08	60	3 (2019) Nature Nanotechnology ⁷
LiFePO4- Li-Cu-CNF -Li	rt	4.5-7.5	0.15	200	4 (2021) Nature ⁸
LiFePO4- PEO/LSTZ	45	3.5	0.15	350	5 (2019)

-Li					Proceedings of the National Academy of Sciences ⁹
LiFePO4- CPE-05MC -Li	55	5	0.5	100	6 (2021) Journal of the American Chemical Society ¹⁰
LiFePO4- HSE-EMIM-PP13 -Li	rt	6.5	0.18	50	7 (2022) Nature Nanotechnology ¹¹
LiFePO4- IPEs -Li	rt	10.3	0.83	350	This work
LiFePO4- IPEs -Li	50	10.3	3.32	350	This work

Comment #3: While the idea behind the simultaneous classification/prediction machine learning algorithm is brilliant, I am not convinced that this work will be of great significance for the field. The main reason for this is the absence of adequate validation of their predictions, which makes then their classification ambiguous. The authors are using a machine learning algorithm to predict the properties of interest in their system (here conductivity and electrochemical window), which however is at no point compared to literature data for the same compounds. I understand that since the studied dataset comes as a result of ion combinations, some of those ionic liquids are not commercially available, or maybe they have never been reported at all. But a comparison on a selected subset of ionic liquids is crucial.

Response: We thank the reviewer for raising these concerns. To validate our prediction results, we compared the predicted ionic conductivity of all the 992 unique liquid ILs in this work to the experimental results stored in ILthermo database. The ILthermo Database contains 523 unique ILs with measured ionic conductivity values. As shown in **Figure R1**, though only 17 ILs are overlapped between our test data and the ILThermo Database at 25 °C, we observe that there is high consistency between the predicted results and the ILthermo results. The R^2 value is 0.76. The mean absolute error (MAE) is 2.03 mS cm⁻¹. Thus we conclude that the model can perform well in predicting the ionic conductivity of ILs.

Figure R1. Comparison of the predicted σ to literature reported σ of the overlapped 17 ILs between the test dataset and the NIST ILThermo Database at 25 °C.

The electrochemical window (ECW) is another popular topic as discussed in the literatures. The ECW value in this manuscript is calculated based on the HOMO/LUMO theory rather than predicted values from the ML model. We note that the ECW is not provided by the ILthermal Database. Thus, we directly compare the calculated ECW with the results scrapped from IoLiTec. In Figure R2, the mean absolute error (MAE) between the calculated results and the experimental results are < 1.1 V. As we know, it is still challenging to estimate ECW for ILs accurately in the field.¹² Besides, the measured ECW values are highly dependent on experimental conditions, thus we believe that the accuracy of the calculated ECW is overall satisfactory and of significant reference to the field.

Figure R2. Comparison of ECW based on IoLiTec to ECW based on HOMO/LUMO theory for the cation(blue) and anion(red) types, correspondingly.

Modifications to the manuscript and supplementary information:

- (1) We added Figure R1 and Figure R2 in the manuscript as Figure 3d and Figure 2 c,d.
- (2) We added more discussion about the validation of predicted ionic conductivity (*Page 13, Line 238 -245*) and calculated ECW (*Page 10, Line 177 – 185*) in the main manuscript.

Comment #4: Regarding the experimental data, from a pool of 40 recommended ionic liquids the authors select 4 to use in their fuel cells. However, I believe that the selection of ionic liquids is not in agreement with the scope of the work. All 4 selected ionic liquids are commercially available and I believe that they were chosen so that the authors avoid synthesising their own compounds. The whole point of this work is to discover an optimised selection of ions for this application, but then the authors decide to work with 1-ethyl-3-methylimidazolium triflate and tetrafluoroborate, which are ‘old’ ionic liquids and they have been used since the 1990’s for electrochemical applications. I believe that as a proof of concept they should have tried at least one ‘unconventional’ combination of ions from their list. The results they obtain on their fuel cells are indeed very promising, but for an expert on the ionic liquid electrochemistry field it is not surprising or unexpected.

Response: We thank the reviewer for mentioning these concerns. The ultimate goal of this work is to facilitate the screening of ILs for IPEs with excellent performance through ML and finally develop a class of IPEs with promising long cycling and fast charge/discharge performance for Li metal battery **instead of fuel cells**. (This is also applied to Comment #13) The selected 4 ILs are all hydrophilic, which is required to form homogeneous and high modulus films with PBDT. Specifically, the excellent performance exhibited by IPEs in this work is not only determined by ionic liquids, but also the liquid crystalline polyelectrolyte PBDT. PBDT is fully hydrophilic, and it has been indicated that PBDT aqueous solution can be ion exchange with hydrophilic IL to form ionogel. However, there is no guideline for selection of ILs previously. It is observed that PBDT can offer mechanical integrity to suppress lithium dendrites, meanwhile significantly reduces the interfacial resistance and improves the charge-transfer kinetics.

As suggested by the reviewer, we also successfully tried one unconventional ionic liquid diethylmethylsulfonium bis(trifluoromethylsulfonyl)imide [Dems][TFSI] from the final recommendation list to fabricate the IPE by changing the solvent from H₂O to H₂O/DMF mixture, we found that this “unconventional” IL is also very promising. We added this IL in the Figure 4 of the manuscript as shown below in **Figure R3**. The detailed experimental performance is summarized in **Figure R4**. We are still conducting in depth investigation on this IL for future work. There are many interesting ILs in the final recommendation list. However, we cannot cover all the combinations experimentally. Above all, we believe this screening method is significant and inspiring for related research areas.

Figure R3. We added the new IL data in Figure 4 of the main manuscript.

Figure R4. (a) Cell voltage versus time for a symmetric Li||Li cell at current densities (J) from 0.1 to 6 mA·cm⁻² with changes in J every 10 cycles at room temperature (each cycle lasts 1 h). (b) Cycling performance of Li|IPEs|LiFePO₄ cell at 0.5C (0.83 mA cm⁻²) at RT. The blue circles show the specific discharge capacity as a function of the increasing cycle number. The black circles display the CE for each cycle correspondingly. (c) The voltage-capacity profiles for the main cycles in (d).

Modifications to the manuscript and supplementary information:

- (1) We updated the Figure 4 in the manuscript with Figure R3 by inserting the new results based on IL Dems TFSI.
- (2) We added Figure R4 in the Supplementary Fig. 6 to support the efficiency of this IL screening workflow.

Comment #5: Does the work support the conclusions and claims, or is additional evidence needed? The experimental investigation is thorough and their results well-explained. The prediction and classification algorithm part though needs further work and validation in order to prove that the algorithm works properly. As I mentioned in the previous section, there is no cross-reference for the prediction accuracy of the calculated properties. This should definitely be amended before publishing this article.

Response: Thanks again for these important suggestions. We believe we have included the answer for these comments in our responses to previous Comments. We have

included the cross-reference in our manuscript. Please also refer to our responses to Comment #3.

Comment #6: Are there any flaws in the data analysis, interpretation and conclusions? - Do these prohibit publication or require revision? During the ‘supervised learning’ section the authors introduce an additional filter to validate their data, by calculating the interaction energy between the anions and the cations. Figure 3c and Supplementary Table 1 show the results of these Energy calculations. To my opinion these calculations are problematic and they need revisiting. According to their formula, the energy values should be negative because the energy of the ion pair ($E[+][-]$) should be more negative than the individual energies of the ions. However, all the energies that are shown in Supplementary Table 1 are positive. The values of $E[+][-]$, $E+$ and $E-$ should also be shown in the ESI, for comparison reasons. Moreover, from my experience, the scale of those energies is unexpected, as I would expect the interaction energies in the range of minus a few hundreds kJ/mol. Depending on the geometry optimization process the interaction energy can differ, so the graph shown in Figure 3c is expected to have an error bar of ± 50 kJ/mol, which makes it impossible to extract certain results from it, as the two box plots mostly overlap. Before the article is considered for publication, the authors need to revisit their calculations, explain properly their calculation methodology and answer questions such as what convergence criteria they use, specify grid accuracy, did they check for absence of imaginary frequencies etc?

Response: We greatly appreciate the reviewer for indicating this error in the manuscript. Yes, we totally agree that the interaction/binding energy between the cation and anion should be negative. We revisit the calculation and append the corrected results and calculation details below. We initially optimize the structure of the cations and anions separately, then the cation-anion pairs are further optimized based on the optimized structure of the cations and anions. The geometry optimization and frequency check are completed with B3LYP/6-311g** with dispersion correction of DFT-D3. The energy calculations are based on M062X/6-311+G(2d, p). The convergence criteria are set by the Gaussian 09 software with default values. (Note, here we choose the commercially available Gaussian instead of Psi4, because Gaussian shows better performance in optimizing the geometry of structure.) Instead of random selection, we select 91 ion pairs, including representative 7 cations and 13 anions from the main cation and anion types in the dataset to validate our prediction results. Among the 91 ion pairs, there are 19 ILs with known phase from IoLiTec. The corresponding average value of the binding energies for the labeled solid and liquid clusters are shown in **Figure R5** and

Table R2 and Table R3. The lower binding energy ($\sim -400 \text{ kJ mol}^{-1}$) of the solid further confirm our conclusion that the binding energy of the solid ion pair usually shows lower binding energy, which means strong interactions between cations and anions.

In terms of the prediction results for the remaining 72 ILs, as shown in **Figure R3**, we divide the predicted results into categories, including liquid and solid-x, where x (x = 1, 2, 3) is the number of ML models with prediction results being in solid phase, thus the higher the number, the larger possibility for the IL being in solid phase. We observe that the predicted liquid cluster showing the highest average binding energy. As x increases, we observe lower average binding energies that further confirms our demonstration.

Modifications to the manuscript and supplementary information:

(1) Figure R5 was added to the main manuscript. More discussion was added to (Page 11, Line 200 - 216) in the manuscript.

(2) Table R2 and Table R3 were added to the Supplementary Table 1 and Table 2.

Figure R5. Blue columns show the binding energy for the ILs with labeled phases. Red columns show the binding energy for the ILs with predicted phases, including liquid and solid-x/3, where x (x = 1, 2, 3) is the number of ML models (SVM, RF, XGB) with prediction results being in solid phase for the ILs.

Table R2. The calculated binding energies for selected cations and anions pairs (19) labeled with phase at RT from IoLiTec.

cation	anion	$E_{\text{opt}}[-]$ (Hartee)	$E_{\text{opt}}[+]$ (Hartee)	$E_{\text{opt}}[+][-]$ (Hartee)	E_{binding} (kJmol ⁻¹)	State IoliTec
1-methyl-1-propylpyrrolidinium	chloride	-460.27	-370.14	-830.55	-385	solid
1-butyl-1-methylpiperidinium	chloride	-460.27	-448.75	-909.16	-389	solid

m						
1-ethyl-3-methylimidazolium	chloride	-460.27	-344.48	-804.90	-403	solid
methylammonium	nitrate	-280.35	-96.19	-376.74	-528	solid
1-ethyl-3-methylimidazolium	nitrate	-280.35	-344.48	-624.98	-398	solid
1-ethyl-3-methylimidazolium	tosylate	-894.82	-344.48	-1239.46	-407	solid
1-methyl-1-propylpyrrolidinium	triflate	-961.56	-370.14	-1331.84	-371	solid
1-butyl-1-methylpiperidinium	triflate	-961.56	-448.75	-1410.44	-361	solid
1-methyl-1-propylpyrrolidinium	tetrafluoroborate	-424.56	-370.14	-794.84	-387	solid
1-butyl-1-methylpiperidinium	tetrafluoroborate	-424.56	-448.75	-873.44	-369	solid
1-ethyl-3-methylimidazolium	bis(fluorosulfonyl)imide	-1351.73	-344.48	-1696.35	-358	liquid
1-ethyl-3-methylimidazolium	methanesulfonate	-663.80	-344.48	-1008.44	-415	liquid
1-ethyl-3-methylimidazolium	ethylsulfate	-778.33	-344.48	-1122.97	-397	liquid
1-ethyl-3-methylimidazolium	thiocyanate	-491.10	-344.48	-835.73	-377	liquid
1-ethyl-3-methylimidazolium	triflate	-961.56	-344.48	-1306.18	-376	liquid
1-ethyl-3-methylimidazolium	tetrafluoroborate	-424.56	-344.48	-769.19	-387	liquid
1-methyl-1-propylpyrrolidinium	dicyanamide	-240.49	-370.14	-610.76	-355	liquid
1-ethyl-3-methylimidazolium	dicyanamide	-240.49	-344.48	-585.11	-369	liquid
1-ethyl-3-methylimidazolium	acetate	-228.51	-344.48	-573.16	-440	liquid

Table R3. The calculated binding energies for selected cations and anions pairs (72) labeled with predicted phases at RT.

cation	anion	$E_{opt}[-]$ (Hartee)	$E_{opt}[+]$ (Hartee)	$E_{opt}[+][-]$ (Hartee)	$E_{binding}$ (kJmol ⁻¹)	State Predict
1-methyl-1-propylpyrrolidinium	bis(fluorosulfonyl)imide	-1351.73	-370.14	-1722.00	-347	Liquid
1-butyl-1-methylpiperidinium	bis(fluorosulfonyl)imide	-1351.73	-448.75	-1800.61	-342	Liquid
diethylmethylsulfonium	bis(fluorosulfonyl)imide	-1351.73	-556.94	-1908.80	-353	Liquid
1-ethyl-3-methylpyridinium	bis(fluorosulfonyl)imide	-1351.73	-366.54	-1718.39	-337	Liquid
ethyltributylphosphonium	bis(fluorosulfonyl)imide	-1351.73	-893.80	-2245.66	-330	Liquid
ethyltributylphosphonium	nitrate	-280.35	-893.80	-1174.30	-382	Liquid
1-ethyl-3-methylimidazolium	dihydrogen phosphate	-643.63	-344.48	-988.28	-440	Liquid
ethyltributylphosphonium	dihydrogen phosphate	-643.63	-893.80	-1537.60	-435	Liquid
ethyltributylphosphonium	methanesulfonate	-663.80	-893.80	-1557.76	-400	Liquid
1-ethyl-3-methylpyridinium	tosylate	-894.82	-366.54	-1261.51	-391	Liquid
ethyltributylphosphonium	tosylate	-894.82	-893.80	-1788.78	-389	Liquid
methylammonium	ethyl sulfate	-778.33	-96.19	-874.71	-496	Liquid
1-methyl-1-propylpyrrolidinium	ethyl sulfate	-778.33	-370.14	-1148.62	-385	Liquid
1-butyl-1-methylpiperidinium	ethyl sulfate	-778.33	-448.75	-1227.23	-389	Liquid
diethylmethylsulfonium	ethyl sulfate	-778.33	-556.94	-1335.42	-405	Liquid
1-ethyl-3-methylpyridinium	ethyl sulfate	-778.33	-366.54	-1145.02	-393	Liquid
ethyltributylphosphonium	ethyl sulfate	-778.33	-893.80	-1672.28	-384	Liquid
methylammonium	thiocyanate	-491.10	-96.19	-587.47	-466	Liquid
1-methyl-1-	thiocyanate	-491.10	-370.14	-861.38	-376	Liquid

propylpyrrolidinium						
1-butyl-1-methylpiperidinium	thiocyanate	-491.10	-448.75	-939.99	-370	Liquid
diethylmethylsulfonium	thiocyanate	-491.10	-556.94	-1048.18	-383	Liquid
1-ethyl-3-methylpyridinium	thiocyanate	-491.10	-366.54	-857.78	-365	Liquid
ethyltributylphosphonium	thiocyanate	-491.10	-893.80	-1385.04	-354	Liquid
ethyltributylphosphonium	triflate	-961.56	-893.80	-1855.50	-358	Liquid
1-ethyl-3-methylpyridinium	tetrafluoroborate	-424.56	-366.54	-791.24	-376	Liquid
1-ethyl-3-methylimidazolium	tricyanomethanide	-316.69	-344.48	-661.31	-356	Liquid
diethylmethylsulfonium	tricyanomethanide	-316.69	-556.94	-873.75	-331	Liquid
1-ethyl-3-methylpyridinium	tricyanomethanide	-316.69	-366.54	-683.36	-343	Liquid
ethyltributylphosphonium	tricyanomethanide	-316.69	-893.80	-1210.62	-327	Liquid
methylammonium	dicyanamide	-240.49	-96.19	-336.85	-452	Liquid
1-butyl-1-methylpiperidinium	dicyanamide	-240.49	-448.75	-689.37	-355	Liquid
diethylmethylsulfonium	dicyanamide	-240.49	-556.94	-797.56	-356	Liquid
1-ethyl-3-methylpyridinium	dicyanamide	-240.49	-366.54	-607.17	-364	Liquid
ethyltributylphosphonium	dicyanamide	-240.49	-893.80	-1134.43	-349	Liquid
1-methyl-1-propylpyrrolidinium	acetate	-228.51	-370.14	-598.81	-438	Liquid
1-butyl-1-methylpiperidinium	acetate	-228.51	-448.75	-677.41	-420	Liquid
diethylmethylsulfonium	acetate	-228.51	-556.94	-785.61	-438	Liquid
1-ethyl-3-methylpyridinium	acetate	-228.51	-366.54	-595.20	-412	Liquid
ethyltributylphosphonium	acetate	-228.51	-893.80	-1122.47	-416	Liquid

hosphonium						
ethyltributylphosphonium	chloride	-460.27	-893.80	-1354.22	-387	Solid-1
methylammonium	bis(fluorosulfonyl)imide	-1351.73	-96.19	-1448.08	-430	Solid-1
diethylmethylsulfonium	dihydrogen phosphate	-643.63	-556.94	-1200.73	-448	Solid-1
diethylmethylsulfonium	methanesulfonate	-663.80	-556.94	-1220.90	-422	Solid-1
1-ethyl-3-methylpyridinium	methanesulfonate	-663.80	-366.54	-1030.49	-406	Solid-1
1-methyl-1-propylpyrrolidinium	tosylate	-894.82	-370.14	-1265.11	-380	Solid-1
1-butyl-1-methylpiperidinium	tosylate	-894.82	-448.75	-1343.71	-356	Solid-1
diethylmethylsulfonium	tosylate	-894.82	-556.94	-1451.92	-409	Solid-1
diethylmethylsulfonium	triflate	-961.56	-556.94	-1518.63	-376	Solid-1
1-ethyl-3-methylpyridinium	triflate	-961.56	-366.54	-1328.23	-368	Solid-1
1-butyl-1-methylpiperidinium	tricyanomethanide	-316.69	-448.75	-765.56	-329	Solid-1
methylammonium	chloride	-460.27	-96.19	-556.65	-525	Solid-2
1-methyl-1-propylpyrrolidinium	nitrate	-280.35	-370.14	-650.64	-385	Solid-2
1-butyl-1-methylpiperidinium	nitrate	-280.35	-448.75	-729.24	-384	Solid-2
diethylmethylsulfonium	nitrate	-280.35	-556.94	-837.44	-404	Solid-2
1-ethyl-3-methylpyridinium	nitrate	-280.35	-366.54	-647.04	-390	Solid-2
1-ethyl-3-methylpyridinium	dihydrogen phosphate	-643.63	-366.54	-1010.33	-425	Solid-2
1-butyl-1-methylpiperidinium	Methane sulfonate	-663.80	-448.75	-1112.70	-398	Solid-2
methylammonium	tosylate	-894.82	-96.19	-991.20	-501	Solid-2
diethylmethylsulfonium	tetrafluoroborate	-424.56	-556.94	-981.64	-389	Solid-2

methylammonium	tricyanomethanide	-316.69	-96.19	-413.03	-404	Solid-2
1-methyl-1-propylpyrrolidinium	tricyanomethanide	-316.69	-370.14	-686.95	-332	Solid-2
methylammonium	acetate	-228.51	-96.19	-324.93	-623	Solid-2
diethylmethylsulfonium	chloride	-460.27	-556.94	-1017.36	-418	Solid-3
1-ethyl-3-methylpyridinium	chloride	-460.27	-366.54	-826.95	-386	Solid-3
methylammonium	dihydrogen phosphate	-643.63	-96.19	-740.03	-575	Solid-3
1-methyl-1-propylpyrrolidinium	dihydrogen phosphate	-643.63	-370.14	-1013.93	-416	Solid-3
1-butyl-1-methylpiperidinium	dihydrogen phosphate	-643.63	-448.75	-1092.54	-417	Solid-3
methylammonium	Methane sulfonate	-663.80	-96.19	-760.18	-517	Solid-3
1-methyl-1-propylpyrrolidinium	Methane sulfonate	-663.80	-370.14	-1034.09	-393	Solid-3
methylammonium	triflate	-961.56	-96.19	-1057.92	-463	Solid-3
methylammonium	tetrafluoroborate	-424.56	-96.19	-520.92	-471	Solid-3
ethyltributylphosphonium	tetrafluoroborate	-424.56	-893.80	-1318.50	-369	Solid-3

Comment #7: Finally, something that is not so much of a flaw in the data analysis but more a part that needs further clarification is the part about the ion exchange with the ionic liquid electrolyte (rows 232-235). It is not clear to me why the incorporation of Li to the IPE is done with LiFSI and not with Li salt with the corresponding anion as the IPE (LiOTf and LiBF₄ accordingly) and also why the ion exchange happens in [C3mpyr][FSI]? I assume that the authors have tried different solvents and have a reason for performing the ion exchange in another IL and not in a conventional volatile molecular solvent, but this should be clearly explained in the text.

Response: We thank the reviewer for initiating the discussion about the choice of LiFSI and C3mpyrFSI. First, it has been reported that the FSI⁻ anion can decompose upon electroreduction to form a stable solid-electrolyte interphase (SEI) that enables reversible cycling with lithium metal anode, and LiFSI has been reported as a promising electrolyte component.¹³⁻¹⁵ Indeed, we had tried LiTfO as the Li salt for this work, but

LiTfO is almost insoluble in C2mimTfO, C2mimFSI and C3mpyrFSI. In contrast, LiFSI has excellent solubility in C3mpyrFSI, we can formulate the ionic liquid electrolytes with high Li⁺ concentrations (3.2 mol kg⁻¹) fairly easily that ensures the high Li⁺ concentration in obtained IPEs after the ion exchange step. In **Table 4**, we observe that the **absolute** value of the binding energy of LiFSI is significantly smaller than that of LiTfO and LiBF₄, this explains why the solubility of LiFSI in ionic liquids is significantly greater than that of LiTfO.

Table R4. The binding energies of different lithium salts.

Lithium salt	Binding energy (kJ mol ⁻¹)
LiBF ₄	-602.39
LiTfO	-603.08
LiFSI	-512.21

Note: The used theory and basis set is M062X/6-311(+)(2d,p).

We avoid using any conventional volatile solvents in the ion exchange procedure, because the organic solvents might cause potential safety issues in real application of Li batteries. We chose C3mpyrFSI as the ionic liquid electrolyte solvent in the ion exchange procedure for two reasons: First, LiFSI has excellent solubility in C3mpyrFSI as mentioned previously, whereas LiFSI is almost insoluble in C2mimTfO and C2mimBF₄. Second, C3mpyrFSI has a wide electrochemical window (5.4 V) and high ionic conductivity (9.1 mS cm⁻¹) simultaneously, which is undoubtedly advantageous for the preparation of IPEs with excellent performance. (Values from Solvionic)

Modifications to the manuscript and supplementary information:

- (1) We added Table R4 and more discussion about the solubility of Li salts in the Supplementary Note 1.
- (2) We added the explanation about the choice of LiFSI and C3mpyrFSI in the manuscript (*Page 4-5, Line 80-89*).

Comment #8: Is the methodology sound? Does the work meet the expected standards in your field? Generally, the methodology of the work follows a reasonable pipeline, although some gaps exist (they are extensively discussed in the previous sections). These gaps need revising before considering this work for publication.

Response: Thanks again for these comments. After the careful validation and detailed explanation of the models included in our responses to previous Comments. We believe that this methodology is suitable and reliable.

Comment #9: I am not convinced where the novelty and the importance of this work lies. There are many published articles on both ML algorithms for property prediction and conductive ionogels. I think that the authors need to add a paragraph (probably in the conclusions) highlighting the novelty of their work, what are the advantages of their algorithm compared to others and why their methodology/IPEs is superior to others reported in the literature.

Response: Thanks for this suggestion. We have added the following paragraph to conclusion of the manuscript.

Modifications to the manuscript and supplementary information:

(1) We added the following paragraph (*Page 25, Line 409 – 431*) in the end of the manuscript to emphasis the novelty and importance of this work from the two perspectives.

“In summary, we have described a machine learning-guided screening protocol to filter promising ILs with high ionic conductivity and wide electrochemical window for preparation of IPEs in LMBs. In terms of the machine learning model, through the unique design of this screening steps based on a combination of unsupervised learning and multistep supervised learning. This comprehensive approach is essential to improve the efficiency to target promising ILs for practical applications. Compared to previous literatures, instead of focusing on individual properties, for example, melting point, viscosity and ionic conductivity, we first combine the factor of ionic conductivity with the electrochemical window as the guidelines for selection of battery electrolytes. This novel conceptual design is also insightful and can be easily applied for related research areas. In addition, though there is plenty of published work based on ML and ionic liquids. It is still difficult to predict the ionic conductivity for ILs accurately because of the data scarcity issues. This work only focuses on unique and commercially available cations and anions from IoLiTec company instead of widely used and scattered NIST ILThermo database. This helps the research work better aligning with the commercially available products, we believe this is also significant for practical research and new materials design in the future. In terms of the electrolyte material development and performance evaluation, the promising experimental results reported in this work

represent the performance of the state-of-the-art polymer electrolytes for Li metal batteries. We further confirm the rigid-rod liquid crystalline polyelectrolyte PBDT as an essential polymer matrix to develop a series of solid-state polymer electrolytes with extremely high CE and excellent fast charge and discharge performance at high temperature. PBDT rods can serve as the assembly templates not only offering mechanical integrity, but also endowing nanoscale structuring in the composite, ensuring the fast Li⁺ transportation.”

Comment #10: Is there enough detail provided in the methods for the work to be reproduced? Yes, apart from the quantum chemistry calculations discussed above, the rest of the methodology is adequately explained and can be reproduced. Regarding the ML algorithm, and while not being an expert on the field, I believe that the algorithm is sound and reasonable. I am not convinced that one could reproduce the algorithm just by reading the article’s methodology section, since there are several gaps on the actual coding, but since the code is available on GitHub one could use the pre-made code.

Response: We thank the review for raising this concern. We added more details for the ML models in the experimental section of the manuscript. We also update the GitHub link for the projects. The class object called ILP can be reused and reproduced for future research.

Modifications to the manuscript and supplementary information:

- (1) We added more details about the ML algorithm (*Page 26, Line 400 – 454*) in the manuscript.
- (2) The updated GitHub link is <https://github.com/wangyingxie/ILP>

Comment #11: There are several grammatical/syntax errors throughout the text, language needs to be corrected.

Response: Thanks for raising these concerns. We have scanned the manuscript again and corrected corresponding the grammatical/syntax errors in the manuscript and the supplementary information.

Comment #12: The company selling the ionic liquids is IoLiTec (in the text it is written as IoLiTech)

Response: We thank the reviewer for pointing out the errors of this work. We have corrected this mistake correspondingly.

Modifications to the manuscript and supplementary information:

(1) We corrected 'IoLiTech' to 'IoLiTec' and other related errors in the manuscript.

Comment #13: The authors refer to their setup as full cell in several points in the text. I believe it is supposed to be fuel cell.

Response: We would like to first appreciate the reviewer for raising this point. Please allow us to clarify this confusion, the IPEs prepared in this work were applied in Li metal batteries rather than fuel cells. The full cell in the manuscript refers to the cell setup of Li|IPES|LiFePO₄, where Li metal is anode and LiFePO₄ is the cathode. Some of the protic ionic liquids, for example the [Dema⁺] based ionic liquids can be potentially employed for fuel cell applications, which are under investigation in our group.

Comment #14: The authors need to amend their abbreviations for their ionic liquids. For example 1-ethyl-3-methylimidazolium triflate, here it is abbreviated as C2mimTFO, the established abbreviation for that is either [Emim][OTf] or [C2C1im][OTf]. The authors need to correct their ILs abbreviations according to the literature.

Response: We totally agree with the established abbreviations proposed by the reviewers for ionic liquids, but in order to keep the abbreviation for ionic liquids consistent with our published work^{16,17} and other literatures¹⁸⁻²⁰, we decided to retain our abbreviations for this ionic liquid in this work. In addition, we have given the full name of the ionic liquid in the manuscript, so we think that the abbreviations for ionic liquids in this work will not cause confusion to the readers.

Comment #15: Figure 4. The embedded photos are too small for anyone to see them properly.

Response: We adjusted all the figures based on the requirements of the journal to ensure the readability and quality of the figures.

Comment #16: Row 52. ‘Large population of ion pairs’ is wrong. It should be a large population of IL candidates.

Response: We greatly appreciate the reviewer’s comment that improves our work a lot.

Modifications to the manuscript and supplementary information:

- (1) We corrected ‘a large population of ion pairs’ to ‘a large population of IL candidates’ in updated row 52.

Comment #17: Row 52. Here they use the abbreviation ML but they define it later in text.

Response: Thanks for this comment. We have modified the corresponding error in the manuscript.

Modifications to the manuscript and supplementary information:

- (1) We defined ML in row 51 and we also removed the definition of ML in row 53.

Comment #18: Row 181. Geometric not gyometric.

Response: Thanks for pointing out this error. We have made corresponding corrections in the manuscript.

Modifications to the manuscript and supplementary information:

- (1) We modified ‘gyometric’ to ‘geometric’ in row 181. (updated *Line 229*)

Comment #19: Row 232. If the authors claim that their films are mechanically strong, they need to perform a tensile stress analysis.

Response: Thanks for this important suggestion. We have supplemented the stress-strain tests for the IPEs with 10% PBDT and C2mimTfO. As shown in **Figure R6a**, the Yield strength and Young’s modulus of the membrane are 6.21 MPa and 300 MPa (the slope of the linear portion on the curve) correspondingly. The Young’s modulus is ~ 3000 times higher than that of PEO-based electrolyte (0.1 MPa)^{21,22}. In **Figure R6b**, the dynamic mechanical analysis (DMA) results show that this membrane maintains a

high modulus >200 MPa from -50 to 300 °C, which ensures the safety and stability of this material as applied to real devices.

Figure R6. (a) The stress-strain curve for IPE with 10% PBDT with C2mim TfO. (b) The corresponding DMA curve for IPE with 10% PBDT with C2mim TfO from -50 to 300 °C.

Modifications to the manuscript and supplementary information:

(1) We added Figure R6 to the Supplementary Fig. 3.

Comment #20: Row 350. The Machine Learning methodology needs to be more detailed in order to be more reproducible.

Response: Thanks for this comment.

Modifications to the manuscript and supplementary information:

(1) We added more details about the ML algorithm (*Page 26, Line 400 – 454*) in the manuscript.

Comment #21: Row 360. The quantum chemistry calculations methodology needs more details to be reproducible.

Response: Thanks for this comment.

Modifications to the manuscript and supplementary information:

(1) We added the calculation details about the quantum chemistry calculation in the experimental section. (*Page 27, Line 455 – 467*)

Comment #22: Rows 365-373. The purity/water content of the purchased ionic liquids needs to be stated. Is Canrd in row 373 correct?

Response: We thank the reviewer for these comments. The purity of all the ionic liquids purchased from IoLiTec are > 99%. All Ionic liquids were further dried in vacuum before moving to glove box. The LiFePO₄ cathode with area loading of 10.3 mg cm⁻² and Cu foil were sourced from Canrd New Energy Technology Co.,Ltd.

Modifications to the manuscript and supplementary information:

- (1) We replaced the “Canrd” mentioned in (*Page 28, Line 478*) with the company’s full name.
- (2) We added the purity of all the ionic liquids purchased from IoLiTec in *Line 472-477*.

Comment #23: Rows 389. The thickness/diameters of the electrodes need to be stated.

Response: We thank the reviewer for this important suggestion. The thickness of the Li metal counter electrode used in cyclic voltammetry experiment is 180 μm and the diameter is 5mm. The thickness/diameter dimension of the working electrodes of LiFePO₄ is 80 μm/4mm and Cu is 10 μm/4mm.

Modifications to the manuscript and supplementary information:

- (1) We added the thickness and diameter of the electrodes in (*Page 29, Line 495-496, 499*).

Comment #24: Row 534. The github link is not working.

Response: We thank the reviewer for pointing out the issue, and we have updated the working GitHub link in the manuscript. The updated link is <https://github.com/wangyingxie/ILP>

Response to Reviewer #2's Comments

Statement: In this manuscript, machine learning approaches are applied to discover ionic liquids with high ionic conductivity and electrochemical window. Some of the top candidates are blended with a polymer and a Li salt. Experiments are conducted to assess the performance of the ionic polymer electrolyte. Given the tremendous interest in Li-ion batteries and favorable attributes of ionic liquids as electrolytes, the topic of the manuscript has broad appeal. Developing electrolytes based on ionic liquids requires careful tuning of properties such as ionic conductivity, electrochemical window, viscosity etc., which are probed in this manuscript. Furthermore, machine learning approaches are becoming mainstream in the field of ionic liquids, so the manuscript holds special interest. Although the topic of the manuscript is of great importance, there are several aspects of the manuscripts which deserve attention and substantial revisions.

Response: We appreciate Reviewer #2's inspiring comments. The manuscript has been modified carefully according to the reviewer's comments. The point-to-point responses are attached as follow.

Comment #1: Very few details on the development of machine learning models are provided. I also find the development of the three machine models rather problematic as the data set is not split into a training and test data set making it difficult to assess the predictive capability of the model. This is especially concerning as it has been pointed out in the manuscript that the models tend to overfit due to over-representation of one or more cation/anion types.

Response: We thank the reviewer for these comments. We added more details about the machine learning models in the experimental section. For the data splitting question, we employed the 5-fold cross validation for the three machine learning models to prevent the potential overfitting. Cross validation is a resampling method that uses different portions of the data to test and train the model. The 5-fold cross validation will split the data into 5 parts, every time 4 parts of the data are used to train the model, the remaining 1 (20% percentage of the data) is used for the validation dataset. Thus, the performance shown in **Table 1** in the manuscript is the average of the 5-fold cross-validation accuracy. In terms of the overfitting issue in other literatures, the reported high accuracy usually originates from the duplicate data points. For example, the ILthermo Database contains properties for diverse IL based on previous literatures, which contains duplicate datapoints for individual IL at varying temperatures. Especially when the model is mainly based on the geometric structure of the cations and

anions, datapoints for same ILs at varying temperatures will artificially increase the training efficiency of the model and cause the issue of overfitting.^{1,2} In this work, to prevent this issue, every ionic liquid in the dataset is unique by permutation of the 74 cations and 30 anions.

Modifications to the manuscript and supplementary information:

(1) We added more details about the ML algorithm (*Page 26, Line 400 – 454*) in the manuscript.

Comment #2: RDKit generates a large number of molecular descriptors for a given ionic liquid. Please include a description of how the total number of features was reduced to 60. Also specify if the cation and anion in an ionic liquid pair were modeled with different set of descriptors.

Response: Thanks for this insightful comment. Basically, we try to include every structural property of the cations and anions. Yes, there is very large number of features provided by RDKit. We start with the default features based on `rdkit.Chem.Descriptors` module (10) and `rdkit.Chem.Descriptors3D` module (10) of the cation, anion and cation-anion pair provided by RDKit. These two modules are representative and contain detailed molecular and geometric properties of the molecules. **The cation and anion in an ionic liquid pair were modeled with same set of descriptors based on RDKit.** We agree that feature engineering is very important for machine learning model. The multicollinearity issue is very apparent in regular linear regression models. However, for SVM, RF and XGboosting models, the multicollinearity will not influence the accuracy of the model. We have tried to simplify the number of features by using Variance Inflation Factor (VIF) methods. However the performance of the model will decreased by removing some of the features. Thus, we decide to keep all of the predetermined 60 features in the final model. On the contrary, we can include more structural features provided by RDKit. However, more features will only increase the complexity and multicollinearity of the model without increasing the accuracy of the model, because the higher order structural features are usually the calculated results based on the variables we already included in the model.

Modifications to the manuscript and supplementary information:

(1) We added more details about the ML algorithm (*Page 26, Line 400 – 454*) in the manuscript. Meanwhile, the updated Github Link will show details about the ML workflow and ensure the reproducibility of this work.

Comment #3: Provide a reasoning for the basis set selection. Please include the level of theory for electronic structure calculations.

Response: Thanks for this question. We employed the Hartee-Fork (HF) theory along with the basis set of 6-311+G** to optimize the geometric structure and followed with the calculation of the energy, the highest occupied molecular orbital energy (E_{HOMO}), the lowest unoccupied molecular orbital energy (E_{LUMO}) and the molecular dipole moment for the cations and anions separately. We also tried to use the popular theory B3LYP to optimize and calculate the energy, E_{HOMO} and E_{LUMO} . However, the obtained ECW values are unreasonable. As reported in literatures,²³ the calculation for ECW based on HOMO/LUMO method heavily relies on the Koopman's theory, which can be well applied with the HF theory; whereas the theory cannot be compatible with the B3LYP theory. The basis set with 6-311+G** usually show enough accuracy according to previous literatures. Overall, the selection of theory and basis set is based on the literatures and the tradeoff between accuracy and cost.

Modifications to the manuscript and supplementary information:

- (1) We added the calculation details about the quantum chemistry calculation in the experimental section. (*Page 27, Line 455 – 467*)
- (2) We added more discussion in the main manuscript (*Page5-6, Line 104-109*).

Comment #3: Why was a particular threshold of ionic conductivity and ECW selected?

Response: The selection of the ionic conductivity and ECW is tunable and can be predetermined based on the specific requirements of practical applications. It is well known that the ionic conductivity of solid-state electrolytes should be $> 1 \text{ mS cm}^{-1}$ to ensure the performance in real devices.²⁴ The IPEs in this work are assembled with a rigid-rod polyelectrolyte, a predetermined lithium salt and ionic liquids (ILs) recommended by machine learning. To guarantee the high conductivity of IPEs, the utilized ILs usually require a slight higher ionic conductivity according to literatures ($>$

5 mS cm⁻¹).^{16,17,25,26} Therefore, 5 mS cm⁻¹ is the ionic conductivity threshold used in this model that should be sufficient for the development of state-of-the-art IPEs.

For the ECW threshold, LiFePO₄ cathode coupled with Li metal anode will display a charging platform at 3.5V (vs Li⁺/Li). Based on the assumption of $V_{CL} \geq V_{Li^+/Li}$, 3.5V will be the minimum threshold value for EC. If the ILs need to match other higher voltage cathodes such as NMC811, LiCoO₂ and LiNiMn₂O₄, the threshold for the ECW can be adjusted appropriately. Here we set the threshold to 4V in the workflow. Here we expand the threshold to 4V in the updated manuscript to ensure the ubiquity.

Modifications to the manuscript and supplementary information:

- (1) We added the more explanations about the thresholds in the manuscript. (*Page 6-7, Line 120-137*)

Comment #4: It is confusing that output properties from models such as ionic conductivity and electrochemical windows are listed as features.

Response: We thank the reviewer for proposing this comment. In the supervised learning, the conductivity and electrochemical window are not listed as features in the machine learning models. We believe that the two features mentioned by the reviewer only exist in the unsupervised learning, the hierarchical clustering method indeed includes the conductivity related factors and the electrochemical window as the input features, which are verified to be independent key properties to filter the ILs for application in Li batteries. Please refer to (*Page 10-11, Line 175 - 189*) for more details.

Comment #5: It is not clear that spherical anions will yield liquids/solid from Figure 3a. In fact, there does not seem to be a correlation between the sphericity of the anions and phase.

Response: We thank the reviewer for proposing this comment. We agree that Figure 3a is not clear to show the dependence, so we reevaluate the correlation between the sphericity of the anions and the phase of the ILs in **Figure R7**. We divided the predicted ILs into the four categories, including liquid and solid-x, where x (x = 1, 2, 3) is the number of ML models with prediction results being in solid phase for the ILs, thus the larger the number, the higher possibility for the IL being in solid phase. To

discover the key features, we start with the features indicated by the feature importance score of the model one by one, finally we observe that the sphericity index of cation and anions will increase with increasing possibility for the cation-anion pair to be solid. The E_{LUMO} of the anion seems like another important feature to determine the phase of the ILs. Indeed, the explanation of the model is quite challenging since the molecular descriptor are highly correlated in the ML models. However, we can still get some basic idea about the important features according to the calculated feature importance from Random Forest and XGBoosting models.

Modifications to the manuscript and supplementary information:

- (1) We added **Figure R7** in the manuscript as **Figure 3b**.
- (2) More discussion was added to the main manuscript (*Page12, Line 219 – 223*)

Figure R7. The key features in the classification of solid/liquid phases of the candidates.

Comment #6: How were top 10 important features determined?

Response: The top 10 importance features can be generated by Random Forest and XGBoosting models directly, the calculated importance score indicates the contribution of each feature in the model. After careful examination of the model, we found that the multicollinearity issue is very severe when we try to explain the importance of the features, though it will not influence the accuracy of the model. The ranking of the importance is fluctuated, but the critical features are usually stable, which give us clue to find the relationship as shown in Figure R7. Thus, to prevent confusion, we decide to remove the Figure 3b from the main manuscript.

Modifications to the manuscript and supplementary information:

(1) We removed original **Figure 3b** in the manuscript.

Comment #7: The results on 20 ILs is not enough to establish the correlation. For 1000 ILs, it is not difficult to conduct quantum calculations. I would recommend at least 20% of the ILs on which these calculations are carried out ensuring that cation types and anions are well represented.

Response: Thanks for this significant suggestion. We have expanded the pool for the calculation. To ensure the representative of the types, we selected 91 ion pairs, including 7 cations and 13 anions from the main cation and anion types in the dataset to validate our prediction results. The results shown in **Figure R5, Table R2 and Table R3** strongly confirms the established correlations. Please refer to **Comment#6 of Reviewer #1** for more details.

(1) Figure R5 was added to the main manuscript. More discussion was added to *(Page 11, Line 200 - 216)* in the manuscript.

(2) Table R2 and Table R3 were added to the Supplementary Table 1 and Table 2.

Comment #8: Why is it necessary to choose hydrophilic ionic liquids for Li-ion batteries? In fact, this property will actually be detrimental to the performance of batteries if water is absorbed.

Response: We appreciate the reviewer for raising this point. As described in the 'Methods' section of the manuscript, the IPEs were prepared by solvent casting the mixture solutions of selected ILs and PBDT. As we know, PBDT can be fully soluble in water instead of any other organic solvents. Therefore, hydrophilic ionic liquids can be easily mixed with PBDT aqueous solutions, to develop homogeneous IPEs with high mechanical strength, meanwhile inhibiting Li dendrite growth and extending the cycle life of Li metal batteries assembled with IPEs. On the contrary, ILEs developed by the hydrophobic ionic liquids usually show phase separation, the developed materials possess a lower mechanical modulus. As the reviewer mentioned, H₂O does have a detrimental effect on the performance of Li metal batteries. For this reason, the developed IPEs were placed in a vacuum oven at 80°C for more than 24h to adequately remove water before assembled in the batteries. The ion exchange process was finished

in an Ar-filled glove box (< 0.01 ppm H_2O). Here we mainly need to measure the H_2O in the dried membrane (10% PBDT C2mim TfO). Here, we used ^1H NMR and DSC to carefully measure the water content in the membrane. As shown in the ^1H NMR spectra in **Figure R8a**, we cannot observe a distinct signal that belongs to H_2O , which usually appears around 4.9 ppm, for the membrane. As shown in the DSC curve in **Figure R8b**, no significant heat absorption peak for H_2O was observed. The excellent battery cycling performance in the manuscript also confirms that the effect of H_2O can be neglected in the IPEs.

Figure R8. (a) ^1H NMR spectra for the membrane. The green dashed line shows the regular position (4.9 ppm) of H_2O peak. (b) DSC curve for the membrane. Notably, we observe no apparent heat absorption peaks above 100 $^{\circ}\text{C}$, which indicates that H_2O molecules were successfully removed after the vacuum drying step.

Modifications to the manuscript and supplementary information:

- (1) Figure R8 was added to Supplementary Fig. 4, which is mentioned at (*Page 16*, *Line 291 – 293*) in the manuscript.

Comment #9: Provide the rationale for using PBDT as a liquid crystalline polyelectrolyte.

Response: We thank the reviewer for this important question. Our previous work has illustrated the importance of PBDT in IPEs (*Nature Materials* 20.9 (2021): 1255–1263). Briefly speaking, the local parallel packing of charged PBDT rods can serve as the assembly templates not only offering mechanical integrity, but also endowing nanoscale structuring in the composite, ensuring the fast Li^+ transpotation¹⁵. In our response to Comment # 2 of Reviewer #1, we conclude a comparison to recent literatures as shown in **Table R1**. Overall, compared to other polymer matrix systems, the IPEs based on PBDT reported in this work outperform from a

comprehensive perspective, including the current density, the cell cycling life and especially the high cathode loading required for practical applications. Furthermore, in our latest study, we found that PBDT at Li metal surface significantly reduces the interfacial resistance and improves the charge-transfer kinetics.

Modifications to the manuscript and supplementary information:

- (1) We added a paragraph (*Page 25, Line 409 – 431*) in the end of the manuscript to emphasize the novelty and importance of this work from the two perspectives.
- (2) We inserted the Table R1 in the main manuscript as Table 2.

Comment #10: The Github link is not provided.

Response: We thank the reviewer for this important suggestion. We have updated the GitHub link in the manuscript. The updated GitHub link is <https://github.com/wangyingxie/ILP>.

Comment #11: Provide a caption for Supplementary Table 2. Include units for ECW and ionic conductivity. Specify the temperature at which ionic conductivity is predicted. For many of the ionic liquids, measured ionic conductivities are available from the NIST ILThermo Database. A comparison must be made between predictions and the measurements.

Response: We thank the reviewer for these comments. As suggested by the reviewer, we have added captions for all Supplementary Tables, and supplemented the units of ECW and ionic conductivity in the corresponding Supplementary Tables. Meanwhile, the ionic conductivity is predicted at 25 °C, which has been added to the manuscript. (*Line 119*) The comparison with the NIST ILThermo databases is also included in the Figure R1 in our responses to Comment#3 of Reviewer #1.

Modifications to the manuscript and supplementary information:

- (1) We added captions for Supplementary Table 1-3, respectively.
- (2) We added the units of ECW and ionic conductivity in Supplementary Table 1-3.
- (3) We added Figure R1 in the manuscript as Figure 3d.
- (4) We added more discussion about the validation of predicted ionic conductivity (*Page 13, Line 238 -245*) in the main manuscript.

References

- 1 Datta, R., Ramprasad, R. & Venkatram, S. Conductivity prediction model for ionic liquids using machine learning. *The Journal of Chemical Physics* **156**, 214505, doi:10.1063/5.0089568 (2022).
- 2 Dhakal, P. & Shah, J. K. A generalized machine learning model for predicting ionic conductivity of ionic liquids. *Molecular Systems Design & Engineering*, doi:10.1039/D2ME00046F (2022).
- 3 Carrera, G. & Aires-de-Sousa, J. Estimation of melting points of pyridinium bromide ionic liquids with decision trees and neural networks. *Green Chemistry* **7**, 20-27, doi:10.1039/B408967G (2005).
- 4 Venkatraman, V. *et al.* Rapid, comprehensive screening of ionic liquids towards sustainable applications. *Sustainable Energy & Fuels* **3**, 2798-2808, doi:10.1039/C9SE00472F (2019).
- 5 Su, Y. *et al.* Rational design of a topological polymeric solid electrolyte for high-performance all-solid-state alkali metal batteries. *Nat Commun* **13**, 4181, doi:10.1038/s41467-022-31792-5 (2022).
- 6 Lin, R. *et al.* Characterization of the structure and chemistry of the solid-electrolyte interface by cryo-EM leads to high-performance solid-state Li-metal batteries. *Nat Nanotechnol* **17**, 768-776, doi:10.1038/s41565-022-01148-7 (2022).
- 7 Wan, J. *et al.* Ultrathin, flexible, solid polymer composite electrolyte enabled with aligned nanoporous host for lithium batteries. *Nat Nanotechnol* **14**, 705-711, doi:10.1038/s41565-019-0465-3 (2019).
- 8 Yang, C. *et al.* Copper-coordinated cellulose ion conductors for solid-state batteries. *Nature* **598**, 590-596, doi:10.1038/s41586-021-03885-6 (2021).
- 9 Xu, H. *et al.* High-performance all-solid-state batteries enabled by salt bonding to perovskite in poly(ethylene oxide). *Proc Natl Acad Sci U S A* **116**, 18815-18821, doi:10.1073/pnas.1907507116 (2019).
- 10 Xu, B. *et al.* Interfacial Chemistry Enables Stable Cycling of All-Solid-State Li Metal Batteries at High Current Densities. *J Am Chem Soc* **143**, 6542-6550, doi:10.1021/jacs.1c00752 (2021).
- 11 Liu, M. *et al.* Improving Li-ion interfacial transport in hybrid solid electrolytes. *Nat Nanotechnol* **17**, 959-967, doi:10.1038/s41565-022-01162-9 (2022).
- 12 Liu, Z. *et al.* Challenges and opportunities for carbon neutrality in China. *Nature Reviews Earth & Environment* **3**, 141-155, doi:10.1038/s43017-021-00244-x (2022).
- 13 Li, T., Zhang, X.-Q., Shi, P. & Zhang, Q. Fluorinated Solid-Electrolyte Interphase in High-Voltage Lithium Metal Batteries. *Joule* **3**, 2647-2661, doi:10.1016/j.joule.2019.09.022 (2019).
- 14 Basile, A., Bhatt, A. I. & O'Mullane, A. P. Stabilizing lithium metal using ionic liquids for long-lived batteries. *Nat Commun* **7**, ncomms11794, doi:10.1038/ncomms11794 (2016).
- 15 Wang, Y. *et al.* Solid-state rigid-rod polymer composite electrolytes with nanocrystalline lithium ion pathways. *Nat Mater* **20**, 1255-1263, doi:10.1038/s41563-021-00995-4 (2021).

- 16 Wang, Y. *et al.* Highly Conductive and Thermally Stable Ion Gels with Tunable Anisotropy and Modulus. *Adv Mater* **28**, 2571-+, doi:10.1002/adma.201505183 (2016).
- 17 Wang, Y. *et al.* Solid-state rigid-rod polymer composite electrolytes with nanocrystalline lithium ion pathways. *Nature Materials* **20**, 1255-1263, doi:10.1038/s41563-021-00995-4 (2021).
- 18 Gao, X., Wu, F., Mariani, A. & Passerini, S. Concentrated Ionic-Liquid-Based Electrolytes for High-Voltage Lithium Batteries with Improved Performance at Room Temperature. **12**, 4185-4193, doi:<https://doi.org/10.1002/cssc.201901739> (2019).
- 19 Yu, T. *et al.* Prediction of the Liquid–Liquid Extraction Properties of Imidazolium-Based Ionic Liquids for the Extraction of Aromatics from Aliphatics. *Journal of chemical information and modeling* **61**, 3376-3385, doi:10.1021/acs.jcim.1c00212 (2021).
- 20 Konieczny, J. K. & Szeferczyk, B. Structure of alkylimidazolium-based ionic liquids at the interface with vacuum and water--a molecular dynamics study. *The journal of physical chemistry. B* **119**, 3795-3807, doi:10.1021/jp510843m (2015).
- 21 Zhou, M. *et al.* Ultrathin Yet Robust Single Lithium-Ion Conducting Quasi-Solid-State Polymer-Brush Electrolytes Enable Ultralong-Life and Dendrite-Free Lithium-Metal Batteries. *Adv Mater* **33**, e2100943, doi:10.1002/adma.202100943 (2021).
- 22 Ma, Y. *et al.* Scalable, Ultrathin, and High-Temperature-Resistant Solid Polymer Electrolytes for Energy-Dense Lithium Metal Batteries. *Advanced Energy Materials* **12**, doi:10.1002/aenm.202103720 (2022).
- 23 Kuusik, I., Kook, M., Pärna, R. & Kisand, V. Ionic Liquid Vapors in Vacuum: Possibility to Derive Anodic Stabilities from DFT and UPS. *ACS omega* **6**, 5255-5265, doi:10.1021/acsomega.0c05369 (2021).
- 24 Zhao, Q., Stalin, S., Zhao, C.-Z. & Archer, L. A. Designing solid-state electrolytes for safe, energy-dense batteries. *Nature Reviews Materials* **5**, 229-252, doi:10.1038/s41578-019-0165-5 (2020).
- 25 Yoon, H., Howlett, P. C., Best, A. S., Forsyth, M. & MacFarlane, D. R. Fast Charge/Discharge of Li Metal Batteries Using an Ionic Liquid Electrolyte. *J Electrochem Soc* **160**, A1629-A1637, doi:10.1149/2.022310jes (2013).
- 26 Yoon, H., Best, A. S., Forsyth, M., MacFarlane, D. R. & Howlett, P. C. Physical properties of high Li-ion content N-propyl-N-methylpyrrolidinium bis(fluorosulfonyl)imide based ionic liquid electrolytes. *Phys Chem Chem Phys* **17**, 4656-4663, doi:10.1039/c4cp05333h (2015).

REVIEWER COMMENTS

Reviewer #1 (Remarks to the Author):

I would like to thank the authors for addressing my comments and providing detailed answers to all of them. Their results are very promising and all the additions have complimented the paper nicely. However, there are still some issues regarding the statistical evaluation of the importance of their classification/prediction algorithm.

In the previous review round (Comment 3) I asked to see a comparison between the predicted properties and experimental measurements available in the literature. The authors kindly did that both for conductivity and ECW. Conductivity is an accurate physical property and their comparison with the experimental data is not great. I understand that there were problems finding experimental data for their ionic liquids (from their 992 predicted ionic liquids, only 17 were on the ILThermo Database), which is a very small percentage to be used as an accurate validation. However, even for those ionic liquids it seems that the accuracy of their prediction is low. The R^2 of the linear predicted-to-experimental conductivity is 0.76, but especially for the low conductivities this leads to huge deviations. It is in the discretion of the Editor to judge whether the 0.76 R^2 factor is satisfactory for publication.

Very accurately the authors noted that the ECW cannot be predicted accurately, as it is significantly influenced by the experimental process – and for that I completely agree. However, the fact that the prediction of ECW is very accurate for some ionic liquids and not for others could be an indication that the model is undertrained or overfitted to specific structures. It would be very interesting for the authors to prove statistically the nature of this deviation. Also, in Figure R2 they should show the number of structures for each ion family studied; for example we see very accurate predictions for pyrrolidinium ILs, while not so much for imidazolium. How many pyrrolidinium ILs and how many imidazolium ILs have been studied?

Similarly, in Figure R5 and Table R2 where the authors show the average binding energy for the solid and the liquid compounds, the authors need to show the statistical significance of their hypothesis. Their hypothesis is that they can associate their calculated binding energy with the physical state of the ionic liquid. If the population of each category (liquid, solid 1, solid 2, solid 3) was the same, this comparison could be done by comparing the standard deviations and see whether they are overlapping. However, since the population of each category is different, the authors need to perform a t-test to prove the statistical significance of their hypothesis.

Reviewer #2 (Remarks to the Author):

In the revised manuscript, authors have addressed all the comments in great detail. It is commendable that they carried out experiments with a not-so-typical ionic liquid to assess the performance of ionic liquid-polymer electrolyte. On the theoretical side, the set of ionic liquids for which quantum calculations were carried has been expanded. Although I still believe that the method concerning selection of features for machine learning has not been described to permit reproduction of results on its own, availability of Github code can alleviate this challenge.

Overall, the manuscripts combines machine learning to identify ionic liquids with suitable ionic conductivity and electrochemical window. Some of the top performing ionic liquids have been mixed with PBDT and promising performance of the solid electrolyte has been reported for Li-ion batteries. I recommend the publication of the manuscript.

Response to Reviewer #1's Comments

Statement: I would like to thank the authors for addressing my comments and providing detailed answers to all of them. Their results are very promising and all the additions have complimented the paper nicely. However, there are still some issues regarding the statistical evaluation of the importance of their classification/prediction algorithm.

Response: We thank the reviewer for complimenting the revisions we made to the manuscript. The corresponding comments proposed by the reviewer again are very insightful. We hope that the following explanation and revision to this manuscript based on the comments will be satisfactory.

Comment #1: In the previous review round (Comment 3) I asked to see a comparison between the predicted properties and experimental measurements available in the literature. The authors kindly did that both for conductivity and ECW. Conductivity is an accurate physical property and their comparison with the experimental data is not great. I understand that there were problems finding experimental data for their ionic liquids (from their 992 predicted ionic liquids, only 17 were on the ILThermo Database), which is a very small percentage to be used as an accurate validation. However, even for those ionic liquids it seems that the accuracy of their prediction is low. The R^2 of the linear predicted-to-experimental conductivity is 0.76, but especially for the low conductivities this leads to huge deviations. It is in the discretion of the Editor to judge whether the 0.76 R^2 factor is satisfactory for publication.

Response: We thank the reviewer for proposing these concerns based on the R^2 factor. We appreciate the chance for us to claim the alternative explanations here. R^2 value is widely used to validate the performance of the models based on the predicted values and experimental values. **It is actually very difficult to directly correlate the performance of the model with the R^2 determinant. The R^2 value is highly dependent on many factors, including the sample size, the data sources and the sample uniqueness, thus we have to evaluate case by case.**

In previous literature,^{1,2} the reported R^2 value can be as high as 0.999, which looks dramatically promising. However, the super high R^2 usually originates from the duplicated input data at various temperatures. As far as we know, most of the related literature using machine learning to predict ionic conductivity is based on the ILThermo

database, which contains 7234 entries of ionic conductivity values at varying temperatures for only 523 unique ILs, the unique cations and anions are high up to 244 and 109 correspondingly, indicating the database is highly sparse. In addition, we can estimate that there are ~ 14 ($7234/523$) records for every unique ionic liquid in the dataset, thus there must be a large number of same ILs appearing both in the training and testing dataset, which will boost the R^2 and lead to the overfitting of the model. Besides, the validation reported in the literature usually relays on the ILThermo database itself. There is seldom validation work using an external database. We found one example,¹ which reports the validation with other literature outside the database they use, though the R^2 values they reported 0.8 and 63% for the two tasks, the value is still based on a wide temperature range from 273 – 303K for same ILs, which will boost the reported R^2 values. **We didn't find reported R^2 in the literature, which compares the unique ILs with external database at a single temperature as we reported here.**

In contrast, we have here is a cross-reference validation between two data sources for unique ILs, including IoLiTec and ILThermo. The IoLiTec contains unique IL datapoints and the measured properties are based on similar measuring conditions, which is more consistent for dataset used for ML models. Besides, the comparison in our work is only based on temperature at 25 °C. The comparison is much stricter and the 0.76 of R^2 factor actually exceed my expectation initially. Imagine if the R^2 is high up to $> 90\%$, the experimental results will suffer an eclipse and become redundant, because we can predict ionic conductivity with enough high accuracy for applications in the future; whereas, the experimental measurements in reality usually display high variation also. For example, when we do the comparison, we note that some ILs among the 18 overlap (instead of 17) in the ILThermo contains more than one record at same temperature (note we use the average values for our validation in Figure 2), as shown in the table as follow, 4 ILs contains more than two records, but the variations between the records is very high, this uncertainty from the database (experimental values) itself will further increase the fluctuation of our validation shown in Figure 2. **Thus, we leave out this invalid value point for 1-butylpyridinium dicyanamide with a huge percentage of difference for the measured values ($> 66\%$), then the R^2 increases to 0.82 and MAE decreases to 1.76.** We update the label and insert the error bar of standard deviation for the four ILs in the figure as follows. **In terms of the data availability, it is difficult for us to increase this number now, since we believe the ILThermo is the biggest database with record literature values for now and we include all the overlaps here without any data bias except the one invalid value we mentioned**

above. Even though there are only 18 overlaps between the two datasets, we find that the model did a nice job to predict IL especially with high ionic conductivity, which is one of our essential targets. Though there is only 18 overlap for validation, we instead obtain a large number of unknown samples to explore in the future. Thus, we think the importance of the relative higher deviation for the low conductivities will be a lower priority in this work. Additionally, the validation of the model is not only limited to the conductivity values, we would like to emphasize again that the uniqueness of the work here is the unique design of this screening steps based on a combination of unsupervised learning and multistep supervised learning.

Last but very important is that the ultimate goal of this work is not simply optimizing the accuracy of the prediction model, but improving the efficiency to screen the suitable ILs for IPEs, which have been verified with excellent performance experimentally. As suggested by the reviewer, based on the recommendation list of the model, we added the performance based on one non-typical IL in our last revision, which further confirms the efficiency of the model. Meanwhile, there are still many ILs on the list will deserve more investigation in the future, which will be very insightful for other researchers in the field.

Above all, we conclude that the model is important and the distinctive R^2 value is insightful to the field. This also indicates that we can pay more attention to the commonly existing bias of database collection and management for future ML investigations.

Modifications to the manuscript and supplementary information:

- (1) We added the following description in the main manuscript about the R^2 factor on Pages 13-14, Lines 249-256.

“As shown in **Fig. 3d**, though only 18 ILs are overlapped between our predicted σ and ILThermo database σ at 25 °C (273.15 K), we observe high consistency between the predicted results and the ILthermo results, especially for those with high σ . The R^2 factor is 0.82, and the mean absolute error (MAE) is 1.76 mS cm⁻¹. More discussions about the R^2 based on this distinctive validation are included in Supplementary Note 7. It is also difficult for us to find similar R^2 validation results in the literature, which compares the unique ILs with external database at a single temperature as we reported here. Overall, we conclude that this model can perform well in predicting the σ of ILs with high σ .”

(2) We updated the new plot in the Figure 2 of the manuscript and added more discussion about the R^2 factor as shown above and the following table in Supplementary Note 7.

Table 1. The overlapped ILs between IoLiTec and ILThermo data sources.

Number	Label	Conductivity (mS cm ⁻¹) 25°C		
		Record 1	Record 2	Record 3
1	1-butyl-3-methylimidazolium trifluoroacetate	3.1	3.32	4.55
2	1-butylpyridinium dicyanamide	8.7	14.8	
3	1-ethyl-3-methylimidazolium methyl sulfate	5.47	6.02	
4	propylammonium acetate	0.43	0.6	
5	1,2-dimethyl-3-propylimidazolium thiocyanate	4.59		
6	1,3-dimethylimidazolium acetate	2.86		
7	1-butyl-2,3-dimethylimidazolium thiocyanate	2.17		
8	1-butyl-3-ethylimidazolium bromide	0.502		
9	1-butyl-3-methylimidazolium dihydrogen phosphate	4.19		
10	1-ethyl-3-methylimidazolium trifluoroacetate	10		
11	1-ethylpyridinium dicyanamide	17.18		
12	1-hexyl-3-methylimidazolium dicyanamide	5.17		
13	1-hexylpyridinium dicyanamide	4.6		
14	1-methyl-3-pentylimidazolium dihydrogen phosphate	2.43		
15	1-methyl-3-propylimidazolium dicyanamide	17.46		
16	1-propylpyridinium dicyanamide	13.08		
17	1-propylpyridinium tetrafluoroborate	4.01		
18	propylammonium formate	3.6		

(d) Comparison of the predicted σ to literature reported σ of the overlapped 17 ILs between the test dataset and the ILThermo Database at 25 °C. The red circle is the invalid point with large experimental uncertainty.

Comment #2: Very accurately the authors noted that the ECW cannot be predicted accurately, as it is significantly influenced by the experimental process – and for that I completely agree. However, the fact that the prediction of ECW is very accurate for some ionic liquids and not for others could be an indication that the model is undertrained or overfitted to specific structures. It would be very interesting for the authors to prove statistically the nature of this deviation. Also, in Figure R2 they should show the number of structures for each ion family studied; for example we see very accurate predictions for pyrrolidinium ILs, while not so much for imidazolium. How many pyrrolidinium ILs and how many imidazolium ILs have been studied?

Response: We thank the reviewer for raising these important questions. We believe there might be some confusion about the method we obtain for the ECW values. The ECW values are not predicted values based on the ML models, but according to the HOMO/LUMO theory calculation results. In terms of the uncertainty for different cation and anion types, it has been discussed in previous literature.^{3,4} Overall, the inconsistency is highly related to the limitation of this theory for estimating particular cation and anion pairs, for example, the C₂mimBF₄ is usually overestimated with a “weak” cation paired with a strong anion. We also emphasize in the manuscript in *Line 313* about the deviation of the BF₄ anion. For the cations, the imidazolium type is also not very accurate, because the description of the top of the valence band for some of the imidazolium-based ILs is not very accurate using the DFT and related approximations, especially for the imidazolium ones with BF₄ anions.⁴ However, the overall trend of the ECW values is reliable and shows enough accuracy for screening of potential IL in this application. Thanks very much for the reviewer’s suggestion about adding the number of studied ILs on the plots. We agree that this is very important for us to see the distribution of the dataset. There are only 47 ionic liquids with measured ECW in IoLiTec. We added the label for each group in the updated plot as below.

Modifications to the manuscript and supplementary information:

- (1) We added a comment on (*Page 10, Lines 182 – 184*) in the manuscript to emphasize this deviation for ECW.

“We observe that the derivations for some cation and anion types are higher. The explanation for the uncertainty in groups like imidazolium and BF₄ is included in Supplementary Note 3.”

- (2) We added the above discussion about the deviation for imidazolium and BF₄ type ILs in Supplementary Note 3 and updated the plots in the manuscript.

Comment #3: Similarly, in Figure R5 and Table R2 where the authors show the average binding energy for the solid and the liquid compounds, the authors need to show the statistical significance of their hypothesis. Their hypothesis is that they can associate their calculated binding energy with the physical state of the ionic liquid. If the population of each category (liquid, solid 1, solid 2, solid 3) was the same, this comparison could be done by comparing the standard deviations and see whether they are overlapping. However, since the population of each category is different, the authors need to perform a t-test to prove the statistical significance of their hypothesis.

Response: We thank the reviewer for these important suggestions. We highly agree that the t-test is a valuable method to further confirm our demonstration. Here, we actually use a combination of **one-way ANOVA** and the **two-sample T-test with equal variance (The equal variance was confirmed with F-test for all T-tests) to confirm our demonstration**. The results for the two tests are appended below. We observe a significant difference between the groups based on the p-value $0.0045 < 0.05$ for the one-way ANOVA. However, the null hypothesis for the one-way ANOVA is that there are at least two pairs that have significant differences for the means. Thus, as shown in table 3, we further performed the t-test for four pairs, including the liquid-solid1/3, liquid-solid2/3 and, liquid-solid 3/3 and liquid-solid-all. As shown in the T-test table, besides the first pair, the other pairs show significant small p-values < 0.05 , thus we reject the null hypothesis is that there is no difference between these groups. Thus, we can conclude that there is a significant difference between the liquid group and groups with two models showing “solid” prediction results. We believe that there will be some overlapping between the groups, because the binding energy is not the only one important factor to determine the phase, many other factors will also influence the final

state of the ILs. Thus, the ML model will play a very important role to combine every factor to give us more reliable results. Similarly, the number in each group has been added to the updated plot in the brackets as well shown below.

Overall, based on the t-test results, we can conclude that the binding energy is a very critical physical property to classify the phase of ILs.

Table 2. ANOVA

Source of Variation	SS	df	MS	F	P-value	F crit
Between Groups	38521.58	3	12840.53	4.77142	0.004459	2.739502
Within Groups	182997.1	68	2691.134			
Total	221518.7	71				

Table 3. The t-test results between the pairs.

t-test	p-val(one_tail)	t_Stat	t_Critical	df
Liquid vs. Solid-1/3	0.299	0.53	1.667	48
Liquid vs. Solid-2/3	0.0057	2.63	1.676	49
Liquid vs. Solid-3/3	0.00047	3.53	1.678	47
Liquid vs. Solid-all	0.002559	2.89	1.667	70

Modifications to the manuscript and supplementary information:

- (1) The above two test table results have been added to Supplementary Note 5. More description has been added to the main manuscript on *Page 12, Line 218 - 224*.

“To validate statistically, we perform both one-way ANOVA and T-test to validate the difference between the liquid cluster as compared to the other three solid groups. Both of the hypothesis testing results indicate significant differences as shown in Supplementary Note 5. The T-test shows more details and indicates significant differences between the liquid and Solid-2/3, Solid-3/3 except for Solid-1/3. Thus, we can conclude that there is a significant difference between the liquid group and groups with more than 2 models showing solid prediction results.”

- (2) Figure 3a has been updated with the number labels for each group in the main manuscript.

Response to Reviewer #2's Comments

Statement: In the revised manuscript, authors have addressed all the comments in great detail. It is commendable that they carried out experiments with a not-so-typical ionic liquid to assess the performance of ionic liquid-polymer electrolyte. On the theoretical side, the set of ionic liquids for which quantum calculations were carried has been expanded. Although I still believe that the method concerning selection of features for machine learning has not been described to permit reproduction of results on its own, availability of Github code can alleviate this challenge.

Overall, the manuscripts combines machine learning to identify ionic liquids with suitable ionic conductivity and electrochemical window. Some of the top performing ionic liquids have been mixed with PBDT and promising performance of the solid electrolyte has been reported for Li-ion batteries. I recommend the publication of the manuscript.

Response: We greatly appreciate Reviewer #2's comments again. We added more description about the selection of features for the ML in the **Method** section of the manuscript, thus ensuring the integrity of the manuscript alone. Additionally, we append the GitHub link again for your reference, which has been available to the public community since our last revision. We expect to see more insight on the website.

Modifications to the manuscript and supplementary information:

(1) We added more details about the selection of ML features on *Page 26, Lines 455-460*.

“Default features based on rdkit.Chem.Descriptors (10) module and rdkit.Chem.Descriptors3D (10) module of the cation, anion and cation-anion pair were obtained by RDKit. These two modules are representative and contain detailed molecular and geometric properties of the molecules. The cation, anion and cation-anion pair were modeled with the same set of descriptors based on RDKit. The remaining 14 features based on Psi4 will be introduced in the quantum chemistry calculation as below.”

(2) The GitHub link is <https://github.com/wangyingxie/ILP>

Reference

- 1 Dhakal, P. & Shah, J. K. A generalized machine learning model for predicting ionic conductivity of ionic liquids. *Molecular Systems Design & Engineering*, doi:10.1039/D2ME00046F (2022).
- 2 Datta, R., Ramprasad, R. & Venkatram, S. Conductivity prediction model for ionic liquids using machine learning. *The Journal of Chemical Physics* **156**, 214505, doi:10.1063/5.0089568 (2022).
- 3 Ong, S. P., Andreussi, O., Wu, Y., Marzari, N. & Ceder, G. Electrochemical Windows of Room-Temperature Ionic Liquids from Molecular Dynamics and Density Functional Theory Calculations. *Chem Mater* **23**, 2979-2986, doi:10.1021/cm200679y (2011).
- 4 Kuusik, I., Kook, M., Pärna, R. & Kisand, V. Ionic Liquid Vapors in Vacuum: Possibility to Derive Anodic Stabilities from DFT and UPS. *ACS omega* **6**, 5255-5265, doi:10.1021/acsomega.0c05369 (2021).

REVIEWERS' COMMENTS

Reviewer #1 (Remarks to the Author):

I thank the authors for replying to my previous comments. I believe this work is now ready for publication.

Response to Reviewer #1's Comments

Statement: I thank the authors for replying to my previous comments. I believe this work is now ready for publication.

Response: We thank the reviewer for this recommendation.